# *Plasmodium*-specific atypical memory B cells are short-lived activated B cells

**Damián Pérez-Mazliah[1], Peter J Gardner[2], Edina Schweighoffer[1], Sarah McLaughlin[1], Caroline Hosking[1], Irene Tumwine[1], Randall S Davis[3,4,5], Alexandre J Potocnik[6], Victor LJ Tybulewicz[1], Jean Langhorne[1]\***

[1]The Francis Crick Institute, London, United Kingdom; [2]MRC National Institute for Medical Research, London, United Kingdom; [3]Department of Medicine, University of Alabama at Birmingham, Birmingham, United States; [4]Department of Microbiology, University of Alabama at Birmingham, Birmingham, United States; [5]Department of Biochemistry and Molecular Genetics, University of Alabama at Birmingham, Birmingham, United States; [6]School of Biological Sciences, The University of Edinburgh, Edinburgh, United Kingdom

**Abstract** A subset of atypical memory B cells accumulates in malaria and several infections, autoimmune disorders and aging in both humans and mice. It has been suggested these cells are exhausted long-lived memory B cells, and their accumulation may contribute to poor acquisition of long-lasting immunity to certain chronic infections, such as malaria and HIV. Here, we generated an immunoglobulin heavy chain knock-in mouse with a BCR that recognizes MSP1 of the rodent malaria parasite, *Plasmodium chabaudi*. In combination with a mosquito-initiated *P. chabaudi* infection, we show that *Plasmodium*-specific atypical memory B cells are short-lived and disappear upon natural resolution of chronic infection. These cells show features of activation, proliferation, DNA replication, and plasmablasts. Our data demonstrate that *Plasmodium*-specific atypical memory B cells are not a subset of long-lived memory B cells, but rather short-lived activated cells, and part of a physiologic ongoing B-cell response.
DOI: https://doi.org/10.7554/eLife.39800.001

**\*For correspondence:**
Jean.Langhorne@crick.ac.uk

**Competing interests:** The authors declare that no competing interests exist.

## Introduction

Atypical memory B cells (AMB) are an unusual B-cell subset detected in both mouse models and humans in the context of certain infections and autoimmune disorders, including HIV, HCV, tuberculosis, malaria, rheumatoid arthritis and systemic lupus erythematosus, and accumulated with age (*Knox et al., 2017b*; *Naradikian et al., 2016a*; *Portugal et al., 2017*; *Rubtsov et al., 2017*). In the context of infections, AMB were first described in HIV-viremic subjects, and termed tissue-like memory B cells, due to their similarity to an FCRL4-expressing memory B-cell subset found in human tonsillar tissues (*Ehrhardt et al., 2005*; *Moir et al., 2008*). In addition to FCRL4, these cells express relatively high levels of other potentially inhibitory receptors including CD22, CD85j, CD85k, LAIR-1, CD72, and PD-1, and show a profile of trafficking receptors including expression of CD11b, CD11c and CXCR3, consistent with migration to inflamed tissues. They are antigen-experienced class-switched B cells, which lack the expression of CD21 and the hallmark human memory B-cell marker CD27. Further studies demonstrated the expression of the transcription factor T-bet and the cytokine IFNγ by these cells, also characteristic of Th1 CD4[+] T cells (*Knox et al., 2017b*; *Obeng-Adjei et al., 2017*; *Portugal et al., 2017*). Due to their poor functional capacity upon in vitro re-stimulation with BCR ligands, AMB were characterized as dysfunctional B cells, and increased frequencies of these cells was proposed to be a consequence of B-cell exhaustion driven by chronic inflammation and stimulation, drawing parallels with T-cell exhaustion during chronic viral infections

(*Moir et al., 2008*; *Portugal et al., 2015*; *Sullivan et al., 2015*). It has been hypothesized that expansion of AMB might contribute to the mechanisms driving autoimmune disorders and deficiencies in acquisition of immunity to chronic infections. However, due to lack of good tools and animal models to analyze antigen-specific atypical B cells in greater depth, many of these concepts remain speculative.

Several studies suggest that AMB might contribute to poor acquisition of long-term immunity to *Plasmodium* infection (*Illingworth et al., 2013*; *Portugal et al., 2015*; *Sullivan et al., 2015*; *Sullivan et al., 2016*; *Weiss et al., 2011*; *Weiss et al., 2009*; *Weiss et al., 2010*). Indeed, some studies demonstrated that in the absence of constant re-exposure, *Plasmodium*-specific serum antibody levels rapidly wane, and full protection from clinical symptoms is lost, suggesting that B-cell memory is functionally impaired (*Portugal et al., 2013*). However, others have reported long-lasting maintenance of *Plasmodium*-specific antibodies and/or memory B cells in settings of differing malaria endemicity, and similar responses are also observed in mouse malaria models (*Dorfman et al., 2005*; *Ndungu et al., 2009*; *Ndungu et al., 2013*; *Ndungu et al., 2012*; *Wipasa et al., 2010*). Moreover, it has been shown that BCRs cloned from *P. falciparum*-specific AMB from malaria-exposed adults encode *P. falciparum*-specific IgG antibodies, which could contribute to *P. falciparum*-specific IgG antibodies in serum (*Muellenbeck et al., 2013*). These authors proposed that *P. falciparum*-specific AMB do not prevent, but rather contribute to the control of *Plasmodium* infection. These apparently contradictory results may reflect the fact that some studies were performed on the general peripheral blood B-cell pool and others focused on *Plasmodium*-specific B cells. In determining a role for these cells in a chronic infection it would be important to follow antigen-specific responses and to distinguish these from non-specific polyclonal B cell activation.

The study of the development of AMB is challenging and requires suitable mouse models, which allow for identification and isolation of antigen-specific B cells that exist often at very low frequency. Here, we generated a knock-in transgenic mouse with a high frequency of B cells specific to the 21 kDa C-terminal fragment of *Plasmodium chabaudi* Merozoite Surface Protein 1 ($MSP1_{21}$), to investigate memory B cells generated following mosquito-transmission of the rodent malaria, *P. chabaudi*. We identified a $CD11b^+CD11c^+FCRL5^{hi}$ subset of $MSP1_{21}$-specific B cells during the chronic infection with phenotypical and transcriptional features strikingly similar to those of human AMB. These AMB disappeared as the infection progressed, leaving a $CD11b^-CD11c^-FCRL5^{hi}$ $MSP1_{21}$-specific B-cell compartment with characteristics of long-lived classical memory B cells ($B_{mem}$) after the resolution of the infection. These short-lived $MSP1_{21}$-specific AMB were also generated in response to immunization, suggesting they may be a normal but transient component of a B-cell response to antigen. In this chronic *P. chabaudi* infection, it appears that AMB require ongoing antigenic stimulation driven by the sub-patent infection to persist, and do not represent a true long-lived 'memory' B cell subset. Moreover, we show that generation of *Plasmodium*-specific AMB does not prevent the generation of *Plasmodium*-specific $B_{mem}$, and does not prevent resolution of the infection.

## Results

### Generation of an immunoglobulin heavy chain knock-in transgenic mouse model to study *Plasmodium*-specific B cell responses

To study Plasmodium-specific B cell responses in a rodent malaria model, we generated an $Igh^{NIMP23/+}$ mouse strain on the C57BL/6J background (Materials and methods and *Figure 1—figure supplement 1*).

The $Igh^{NIMP23/+}$ mice were healthy, with no unusual behavioral or physical characteristics. There were no alterations in total cellularity, pro-B, pre-B, immature B, mature B, total $B220^+CD19^+$ B cells, and plasma cells in the bone marrow (*Figure 1A–B*), and no alterations in number of T1, T2, T3, follicular, marginal zone, germinal center B cells, plasmablasts, plasma cells, and total cellularity in the spleen of $Igh^{NIMP23/+}$ mice (*Figure 1C–D*) (*Sen et al., 1990*; *Young et al., 1994*). Importantly, The $Igh^{NIMP23/+}$ mice had a greatly increased frequency of B cells specific for $MSP1_{21}$ (approximately 60% of the total B-cell compartment), as demonstrated by flow cytometry analysis of splenocytes with a $MSP1_{21}$ fluorescent probe consisting of biotinylated recombinant $MSP1_{21}$ loaded on streptavidin-PE (*Figure 1E–F*). Thus, in this model, a recombinant light chain is not required to bring about

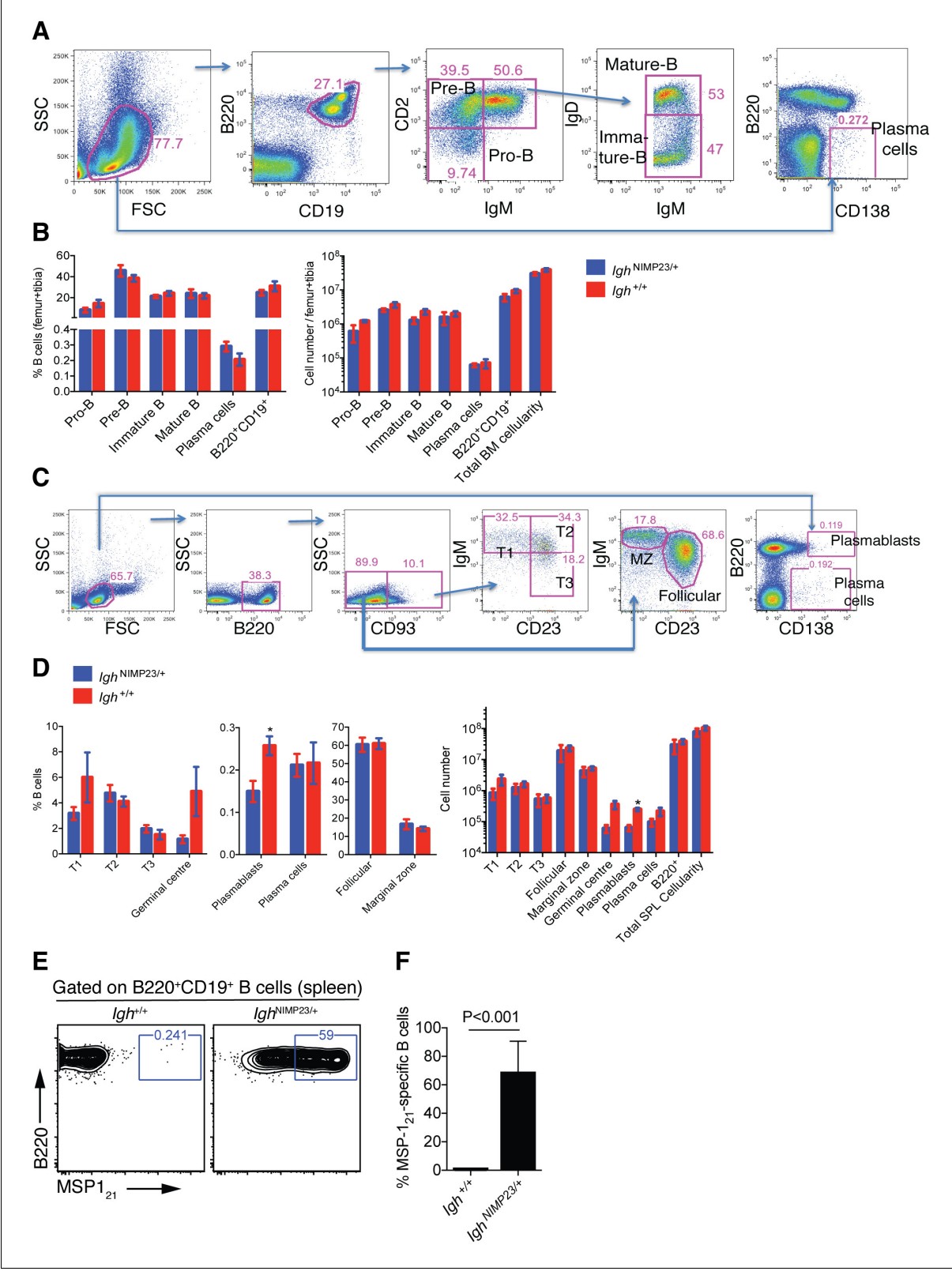

**Figure 1.** Analysis of total bone marrow and splenic B-cell populations in $Igh^{NIMP23/+}$ and $Igh^{+/+}$ littermates. (**A**) Flow cytometry gating strategy to identify different B-cell populations in bone marrow of $Igh^{NIMP23/+}$ mice. Arrows indicate flow of analysis. The same strategy was used for $Igh^{+/+}$ littermates. (**B**) Percentages and numbers of different B-cell populations in bone marrow of $Igh^{NIMP23/+}$ and $Igh^{+/+}$ littermates as defined in (**A**). (**C**) Flow cytometry gating strategy to identify different B-cell populations in spleen of $Igh^{NIMP23/+}$ mice. (**D**) Percentages and numbers of different B-cell

*Figure 1 continued on next page*

*Figure 1 continued*

populations in spleen of $Igh^{NIMP23/+}$ and $Igh^{+/+}$ littermates as defined in (C). Data are representative of two independent experiments with four mice per group. (E) Flow cytometry analysis of B cells obtained from spleen of $Igh^{+/+}$ (left) and $Igh^{NIMP23/+}$ (right) mice stained with anti-B220 and CD19 antibodies in combination with an MSP1$_{21}$ fluorescent probe. The gates show the frequency of B cells specific to MSP1$_{21}$. (F) Frequencies of MSP1$_{21}$-specific splenic B cells in $Igh^{NIMP23/+}$ and wild-type $Igh^{+/+}$ littermate controls (Mann Whitney U test). Data pooled from two independent experiments with 3–5 mice per group. Mann Whitney U test. *p<0.05. Error bars are SEM.

DOI: https://doi.org/10.7554/eLife.39800.002

The following figure supplements are available for figure 1:

**Figure supplement 1.** Generation of $Igh^{NIMP23/+}$knock in mice.

DOI: https://doi.org/10.7554/eLife.39800.003

**Figure supplement 2.** Generation of mixed bone marrow chimera model with reduced precursor frequency of $Igh^{NIMP23/+}$ B cells to study MSP1$_{21}$-specific B cell responses during *P. chabaudi* infection.

DOI: https://doi.org/10.7554/eLife.39800.004

specificity. This suggest that most endogenous light chains will pair with the NIMP23 heavy chain to generate a BCR with detectable binding to MSP1$_{21}$.

## Increase in *Plasmodium*-specific B cells after mosquito transmission of *P. chabaudi*

To investigate B cells in *P. chabaudi* infections, which last several weeks, and to avoid potential problems with activation arising from very high frequencies of MSP1-specific B cells, we reduced the precursor frequency of MSP1$_{21}$-specific B cells to match the natural level expected for antigen-specific B cells more closely, yet still readily detectable by flow cytometry. We generated mixed bone marrow (BM) chimeras by adoptively transferring a mixture of 10% bone marrow from either $Igh^{NIMP23/+}$ or $Igh^{+/+}$ mice (CD45.2$^+$) together with 90% bone marrow from C57BL/6.SJL-$Ptprc^a$ mice (CD45.1$^+$) into sub-lethally irradiated $Rag2^{-/-}$.C57BL/6.SJL-$Ptprc^a$ mice (CD45.1$^+$) to generate NIMP23→$Rag2^{-/-}$ and WT→$Rag2^{-/-}$ bone marrow chimeric mice respectively (*Figure 1—figure supplement 2A–B*). In both types of chimeras, 2–3% of the B cells were CD45.2$^+$ and in NIMP23→$Rag2^{-/-}$ mice approximately 1–2% of the B cells were MSP1$_{21}$-specific (*Figure 1—figure supplement 2C–E*). No MSP1$_{21}$-specific B cells were detected in the control WT→$Rag2^{-/-}$ chimeras (*Figure 1—figure supplement 2D*).

Infection of C57BL/6J *wt* mice with *P. chabaudi* by mosquito bite gives rise to a short (48 hr) pre-erythrocytic infection, followed by an acute blood parasitemia peaking approximately 10d post-transmission. Thereafter, the infection is rapidly controlled, reaching very low parasitemias by 15d post-transmission, with a subsequent prolonged (~90 d), but low-level chronic infection before parasite elimination (*Brugat et al., 2017*; *Spence et al., 2013*). NIMP23→$Rag2^{-/-}$ mice infected with *P. chabaudi* by mosquito bite, showed a similar course of parasitemia to that of control WT→$Rag2^{-/-}$ mice (*Figure 1—figure supplement 2F*), and C57BL/6J *wt* mice (*Brugat et al., 2017*; *Spence et al., 2013*; *Spence et al., 2012*). Importantly, the MSP1$_{21}$-specific $Igh^{NIMP23/+}$ B cells (CD45.2$^+$MSP1$_{21}^+$) in NIMP23→$Rag2^{-/-}$ chimeras showed a robust response to the infection, as demonstrated by a dramatic increase in the proportions and numbers of GL-7$^+$CD38$^{lo}$ germinal centers (GC) and IgG2b$^+$-IgD$^-$ class-switched B cells in the spleen at 35 days post-infection (dpi) (*Figure 1—figure supplement 2G–H*).

Thus, we have generated a mouse model with detectable numbers of functional MSP1$_{21}$-specific B cells capable of responding to *P. chabaudi* infection.

## Generation of *Plasmodium*-specific AMB after mosquito transmission of *P. chabaudi* infection

We investigated whether *Plasmodium*-specific AMB could be identified in mice during a blood-stage *P. chabaudi* infection. We selected a series of mouse homologues to human cell surface markers described on human AMB (*Charles et al., 2011*; *Kardava et al., 2014*; *Kardava et al., 2011*; *Knox et al., 2017a*; *Li et al., 2016*; *Moir et al., 2008*; *Muellenbeck et al., 2013*; *Portugal et al., 2015*; *Russell Knode et al., 2017*; *Sullivan et al., 2015*). Human AMB express CD11b, CD11c, Fc receptor-like (FCRL) 3–5, high levels of CD80, low levels of CD21, and are Ig class-switched. Mouse FCRL5 most closely resembles human FCRL3 and is the only mouse FCRL-family member which

contains both ITIM and ITAM motifs in its cytoplasmic tail (*Davis, 2007*; *Davis et al., 2004*; *Won et al., 2006*; *Zhu et al., 2013*). Therefore, our flow cytometry panel for mouse AMB included antibodies against CD11b, CD11c, FCRL5, CD21, IgD, and also CD80 and CD273 which identify mouse B cells that are antigen-experienced and potentially memory cells (*Anderson et al., 2007*; *Tomayko et al., 2010*; *Zuccarino-Catania et al., 2014*).

We detected an increased number of cells in a distinct CD11b$^+$CD11c$^+$ MSP1$_{21}$-specific B-cell subset at 28-35dpi, in the chronic phase of *P. chabaudi* infection (*Figure 2A–B*). This subset showed several AMB characteristics, including high expression of FCRL5 and low expression of CD21 and IgD (*Figure 2C–E*). In addition, the CD11b$^+$CD11c$^+$ MSP1$_{21}$-specific B-cell subset was enriched with cells expressing CD80 and CD273 (*Figure 2C–E*).

We then explored whether this CD11b$^+$CD11c$^+$ MSP1$_{21}$-specific B cell subset was detected during the memory phase,that is after resolution of the infection. As it takes up to 90 days for a blood-stage *P. chabaudi* infection to be eliminated from C57BL/6J mice (*Achtman et al., 2007*; *Spence et al., 2013*), we measured these responses from 155dpi onwards. Unexpectedly, the numbers of CD11b$^+$CD11c$^+$ MSP1$_{21}$-specific B cells were not significantly higher than background level (*Figure 2A–B*).

These data demonstrate that a mosquito-borne infection with *P. chabaudi* generates *Plasmodium*-specific B cells resembling human AMB. However, these cells do not persist and are not detected above background level after parasite clearance.

## Transcriptome analysis confirms the AMB nature of CD11b$^+$CD11c$^+$ MSP1$_{21}$-specific B cells, and reveals a plasmablast-like signature for this subset

To gain a better understanding of the identity of the CD11b$^+$CD11c$^+$ *Plasmodium*-specific B cell subset, we isolated both CD11b$^+$CD11c$^+$ and CD11b$^-$CD11c$^-$ MSP1$_{21}$-specific B cells from spleens of *P. chabaudi*-infected NIMP23→*Rag2$^{-/-}$* mice (35dpi) (*Figure 3—figure supplement 1*), and MSP1$_{21}$-specific B cells from the spleen of naïve NIMP23→*Rag2$^{-/-}$* mice (*Figure 2A*), by flow cytometric sorting, and performed an mRNAseq transcriptional analysis on the three populations.

We then selected a large series of key genes previously shown to be either up or downregulated on human AMB (*Supplementary file 1*), and explored the expression of their mouse homologues on the three different B-cell subsets we sorted at day 35pi. The transcriptome of CD11b$^+$CD11c$^+$ MSP1$_{21}$-specific B cells highly resembled that of human AMB. A series of hallmark genes upregulated in human AMB were also upregulated in CD11b$^+$CD11c$^+$ MSP1$_{21}$-specific B cells, including IgG (*Ighg2c* and *Ighg2b*), *Cxcr3*, *Tbx21* (T-bet), *Lair1* and *Fcrl5* (*Figure 3*, genes in red boxes, and references in *Supplementary file 1*). In addition, the MSP1$_{21}$-specific CD11b$^+$CD11c$^+$ B-cell subset showed upregulation of *Ifng*, *Aicda*, a large array of inhibitory receptors [including *Pd1*, *Cd72*, *Cd85k*, *Fcgr2b* (CD32b), S*iglece*], antigen-experienced/memory markers [*Cd80*, *Cd86*, *Nt5e* (CD73) and high *Cd38*], and additional class-switched immunoglobulins (i.e. *Igha*, *Ighg1* and *Ighg3*), all of which have been shown to be upregulated on human AMB (*Figure 3C–G*, references in *Supplementary file 1*). These cells also expressed Galectins (*Lgals1* and *Lgals3*), previously implicated in B-cell anergy (*Clark et al., 2007*) (*Figure 3H*), and displayed a pro-apoptotic program (e.g. high expression of *Fasl*, and low expression of *Bcl2*) (*Figure 3I*). Interestingly, in agreement with data on human AMB, MSP1$_{21}$-specific CD11b$^+$CD11c$^+$ B cells showed upregulation of *Mki67* (*Figure 3J*), indicative of proliferation, and had characteristics of plasmablasts and/or plasma cells, including upregulation of *Cd138*, *Prdm1* (Blimp1) and *Xbp1*, and low expression of *Cxcr5*, *Pax5*, and *Bcl6* (*Figure 3B and K*). However, these cells showed low expression of *Irf4* and *S1p1*, suggesting that they may be in a pre-plasmablast or pre-migratory plasma-cell stage (*Kabashima et al., 2006*; *Kallies et al., 2007*) (*Figure 3K*). Finally, and similar to human AMB, CD11b$^+$CD11c$^+$ MSP1$_{21}$-specific B cells showed low expression of *Cd40*, *Cr2* (CD21), *Ms4a1* (CD20), and *Cd24a* (*Figure 3L*).

We ran a Gene Set Enrichment Analysis (GSEA) (*Subramanian et al., 2005*) with a gene list ranked according to their differential expression between MSP1$_{21}$-specific CD11b$^+$CD11c$^+$ AMB sorted from infected mice and MSP1$_{21}$-specific B cells sorted from naïve mice, using gene sets a priori obtained from Reactome (*Fabregat et al., 2018*). Among the gene sets yielding the top 50 significant (fdr <0.001) highest normalized enrichment score (*NES*) we obtained gene sets corresponding to cell cycle, DNA replication, generation/consumption of energy, regulation of apoptosis, activation of NF-κB on B cells, and downstream signaling events of the BCR

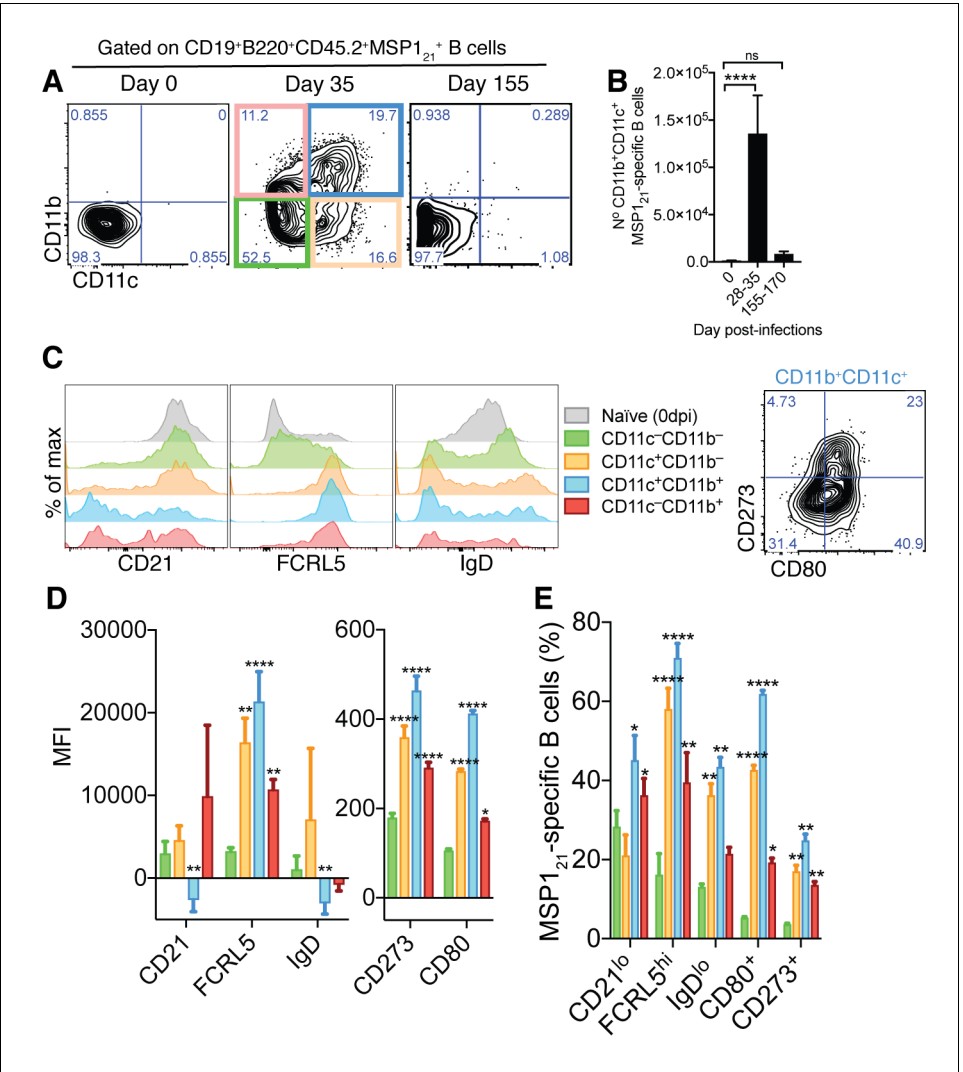

**Figure 2.** Generation of MSP1$_{21}$-specific AMB in response to mosquito transmitted *P.chabaudi* infection. (**A**) Flow cytometry showing differential expression of CD11b and CD11c on splenic MSP1$_{21}$-specific B cells from NIMP23→*Rag2$^{-/-}$* chimeric mice before infection (day 0) and at 35 and 155dpi. (**B**) Numbers of splenic MSP1$_{21}$-specific CD11b$^+$CD11c$^+$ AMB from NIMP23→*Rag2$^{-/-}$* during the course of mosquito transmitted *P. chabaudi* infection. Kruskal-Wallis test vs day 0. ****, p<0.0001 (**C**) Flow cytometry showing expression of CD21/35, FCRL5, IgD, CD273 and CD80 on different subsets of splenic MSP1$_{21}$-specific B cells from NIMP23→*Rag2$^{-/-}$* chimeric mice defined based on CD11b and CD11c expression at 35dpi. (**D**) Geometric mean fluorescence intensity (MFI) of CD21/35, FCRL5, IgD, CD273 and CD80 expression on different subsets of splenic MSP1$_{21}$-specific B cells from NIMP23→*Rag2$^{-/-}$* chimeric mice defined based on CD11b and CD11c expression at 35dpi. (**E**) Frequencies of CD21/35, FCRL5, IgD, CD273 and CD80 positive cells among different subsets of splenic MSP1$_{21}$-specific B cells from NIMP23→*Rag2$^{-/-}$* chimeric mice defined based on CD11b and CD11c expression at 35dpi. Two-way ANOVA vs CD11b$^-$CD11c$^-$ subset. *p<0.05; **p<0.01; ***p<0.001; ****p<0.0001. Error bars are SEM. Data pooled from three independent experiments with 3–5 mice per group.

DOI: https://doi.org/10.7554/eLife.39800.005

(*Supplementary file 2* and *Figure 3—figure supplement 2*). These data further corroborate the activated and proliferative nature of MSP1$_{21}$-specific CD11b$^+$CD11c$^+$ AMB.

Taken together, these data demonstrate that CD11b$^+$CD11c$^+$ MSP1$_{21}$-specific mouse AMB present during the chronic phase of *P. chabaudi* infection are very similar to human AMB described in several chronic infections. In addition, this B-cell subset shows features of activation, proliferation,

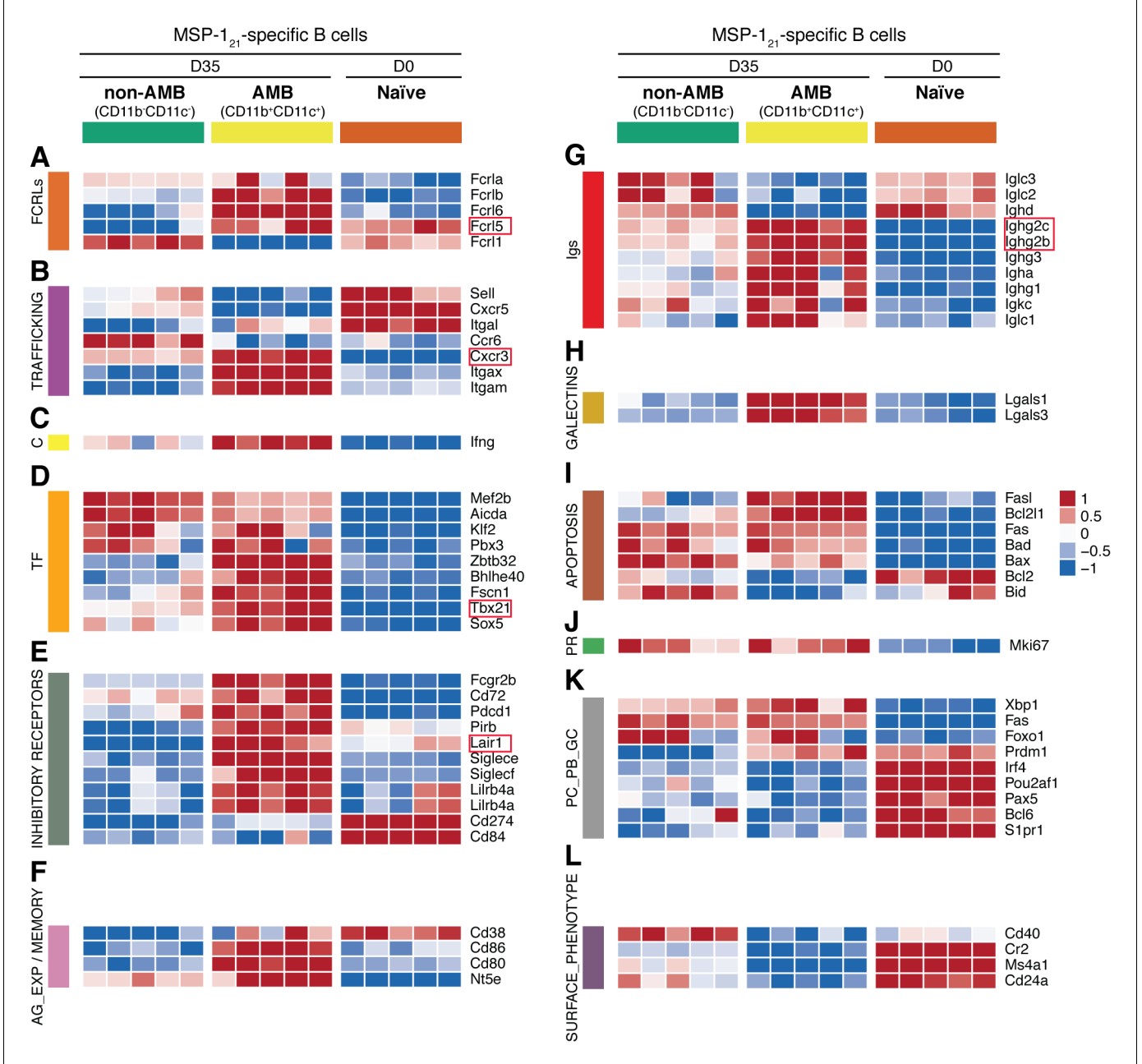

**Figure 3.** Transcriptome analysis of sorted splenic MSP1$_{21}$-specific CD11b$^+$CD11c$^+$AMB. MSP1$_{21}$-specific CD11b$^+$CD11c$^+$ (AMB) and CD11b$^-$CD11c$^-$ B cells were flow cytometry sorted from the spleen of NIMP23→$Rag2^{-/-}$ chimeric mice at 35dpi; MSP1$_{21}$-specific B cells were flow cytometry sorted from the spleen of naïve NIMP23→$Rag2^{-/-}$, and these three B cell populations were submitted to mRNAseq analysis. The heat maps display level of expression of selected individual genes, organized in functional clusters related to (A) Fc receptor like molecules, (B) cell trafficking, (C) cytokines, (D) transcription factors, (E) inhibitory receptors, (F) antigen experience/memory, (G) immunoglobulins, (H) galectins, (I) apoptosis, (J) proliferation, (K) plasma cells/plasmablasts/germinal centers, (L) surface markers. Each column corresponds to data from an individual mouse (n = 5 35 dpi, n = 5 0 dpi).
DOI: https://doi.org/10.7554/eLife.39800.006

The following figure supplements are available for figure 3:

**Figure supplement 1.** Gating strategy for the sorting of splenic MSP1$_{21}$-specific CD11b$^+$CD11c$^+$ AMB and CD11b$^-$CD11c$^-$ B cells at 35dpi.
DOI: https://doi.org/10.7554/eLife.39800.007

**Figure supplement 2.** Profile of the Running *ES* Score and positions of GeneSet members on the rank ordered list for selected Reactome pathway gene sets.
DOI: https://doi.org/10.7554/eLife.39800.008

DNA replication and plasmablasts, resembling previous observations in human AMB (*Muellenbeck et al., 2013*).

## Generation of *Plasmodium*-specific AMB in response to immunization

The occurrence of $CD11b^+CD11c^+$ AMB might be a consequence of aberrant B-cell activation driven exclusively by certain pathogens. Alternatively, they might be part of a normal B-cell response, which is exacerbated by the persistent nature of certain infections. To test whether $CD11b^+CD11c^+FCRL5^+$ AMB could be generated in the absence of persistent infection, we immunized mice with $MSP1_{21}$. A previous report had demonstrated the presence of $CD11b^+CD11c^+Tbet^+$ B-cells 24 hr post-immunization with R848, a TLR7/8 ligand (*Rubtsova et al., 2013*). Therefore, we immunized $Igh^{NIMP23/+}$ mice with R848 together with the antigen $MSP1_{21}$ and looked for the appearance of $MSP1_{21}$-specific $CD11b^+CD11c^+FCRL5^+$ atypical B cells. We observed substantial numbers of $MSP1_{21}$-specific $CD11b^+CD11c^+$ B cells in the spleens of $Igh^{NIMP23/+}$ mice 24 hr post-immunization (*Figure 4A*). These cells expressed increased levels of both FCRL5 and CD80 (*Figure 4B–C*) and did not display GC characteristics (*Figure 4D*), similar to the $MSP1_{21}$-specific $CD11b^+CD11c^+FCRL5^+$ atypical B cells generated following *Plasmodium* infection. The $MSP1_{21}$-specific $CD11b^+CD11c^+$ B cells observed

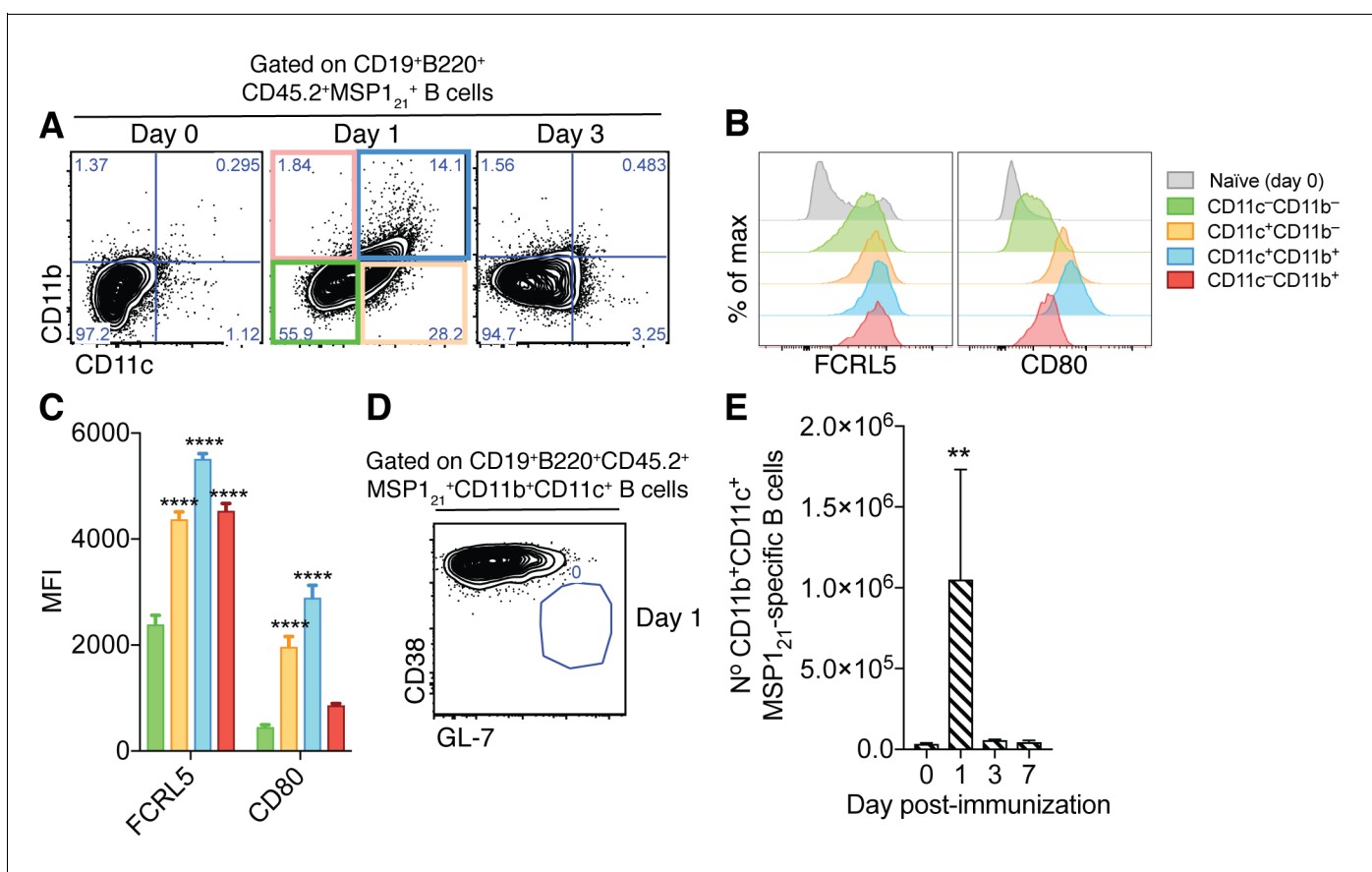

**Figure 4.** Generation of splenic $MSP1_{21}$-specific $CD11b^+CD11c^+$AMB in response to immunization. (**A**) Flow cytometry showing differential expression of CD11b and CD11c on splenic $MSP1_{21}$-specific B cells from $Igh^{NIMP23/+}$ mice before immunization (day 0) and at days 1 and 3 post-immunization with R848 and $MSP1_{21}$. (**B**) Flow cytometry showing expression of FCRL5 and CD80 on different subsets of splenic $MSP1_{21}$-specific B cells from $Igh^{NIMP23/+}$ defined based on CD11b and CD11c expression at day one post-immunization and naïve mice. (**C**) Geometric MFI of FCRL5 and CD80 expression on different subsets of splenic $MSP1_{21}$-specific B cells from $Igh^{NIMP23/+}$ defined based on CD11b and CD11c expression at day one post-immunization. Two-way ANOVA vs $CD11b^-CD11c^-$ subset. ****$p<0.0001$. (**D**) Flow cytometry of CD38 vs GL-7 (GC markers) on $CD11b^+CD11c^+$ $MSP1_{21}$-specific B cells from $Igh^{NIMP23/+}$ at day one post-immunization. (**E**) Numbers of splenic $CD11b^+CD11c^+$ $MSP1_{21}$-specific B cells from $Igh^{NIMP23/+}$ during the course of immunization. Kruskal-Wallis test compared to day 0. **$p<0.01$. Error bars are SEM. Data pooled from three independent experiments with 3–5 mice per group.

DOI: https://doi.org/10.7554/eLife.39800.009

after immunization appeared only transiently, as they could no longer be detected at 3 and 7d post-immunization (*Figure 4A and E*).

These data demonstrate that $MSP1_{21}$-specific $CD11b^+CD11c^+$ AMB with no functional characteristics of memory B cells can be generated independently of the infection and the presence of the pathogen, and that they are short-lived cells.

## *Plasmodium*-specific $CD80^+CD273^+$ $B_{mem}$ are generated and persist after resolution of *P. chabaudi* infection

Identification of mouse $B_{mem}$ by flow cytometry originally relied on detecting B cells that had undergone Ig class-switching from IgM to IgG, and that did not express GC markers (i.e. $IgG^+CD38^{hi}GL-7^{lo}$) (*Lalor et al., 1992*; *Ridderstad and Tarlinton, 1998*). More recently, this set of markers has been extended to include CD80, CD273 (PD-L2) and CD73, with CD273 and CD80 being the most useful to discriminate memory from naïve B cells (*Tomayko et al., 2010*; *Zuccarino-Catania et al., 2014*). In combination, these markers allow the identification of different subsets of switched as well as non-class switched (i.e. $IgM/D^+$) $B_{mem}$. Therefore, we used cell surface expression of CD80 and CD273 on $MSP1_{21}$-specific B cells to identify $B_{mem}$ during and after resolution of P. *chabaudi* infection.

$MSP1_{21}$-specific B cells from spleens of naïve NIMP23→$Rag2^{-/-}$ mixed BM chimeras showed little expression of either CD80 or CD273 (*Figure 5A*). By contrast, $CD80^+CD273^+$, $CD80^+CD273^-$ and $CD80^-CD273^+$ $MSP1_{21}$-specific B cells, both class-switched ($IgM/D^{lo}$) and non-class-switched ($IgM/D^{hi}$), were readily detected above background levels at 28-35dpi in *P. chabaudi*-infected NIMP23→$Rag2^{-/-}$ mixed BM chimeras (*Figure 5B and D*). After resolution of infection (155-170dpi), the numbers of $CD80^+CD273^+$ and $CD80^+CD273^-$ $MSP1_{21}$-specific B cells were either sustained or increased, while the $CD80^-CD273^+$ population decreased, compared with 28-35dpi (*Figure 5A–D*). All three $MSP1_{21}$-specific $B_{mem}$ subsets showed high CD38 expression (*Figure 5E,F*). Importantly, the $MSP1_{21}$-specific GC B cells detected at 28-35dpi did not express either CD80 or CD273, further distinguishing GC from memory and atypical memory *P. chabaudi*-specific B-cell subsets (*Figure 5G*). No $MSP1_{21}$-specific GC B cells were detected above background level after resolution of the infection (*Figure 5H*). Finally, and in accordance with *Figure 3*, approximately 70% of the $MSP1_{21}$-specific $CD11b^+CD11c^+$ AMB observed at days 28-35pi expressed either CD80, CD273 or a combination of both (*Figure 5I*).

These data show that, in contrast to the transient AMB, splenic $CD80^+CD273^+$ and $CD80^+CD273^-$ class-switched and non-class-switched $MSP1_{21}$-specific $B_{mem}$ persist after resolution of *P. chabaudi* infection.

## *Plasmodium*-specific $B_{mem}$ express high levels of FCRL5

As discussed above, no single marker has been described so far that can identify all mouse $B_{mem}$ subsets. Surprisingly, we observed that after resolution of infection (155-170dpi), $MSP1_{21}$-specific B cells expressing different combinations of CD80 and CD273 ($CD80^+CD273^+$, $CD80^-CD273^+$ or $CD80^+CD273^-$) all expressed very high levels of FCRL5, in contrast to $CD80^-CD273^-$ $MSP1_{21}$-specific B cells that express no memory markers at this stage (*Figure 6A*). This suggests that FCRL5 might be a marker for all $B_{mem}$. In order to confirm this, we used unsupervised methods to analyze our multiparameter flow cytometry data. We used PhenoGraph and t-SNE within the Cytofkit package (Materials and methods, Chen et al 2016) to analyze $MSP1_{21}$-specific B cells based on the expression of FCRL5, CD38, IgD, CD80 and CD273 on these cells, as determined by flow cytometry (*Figure 6A and B*). The analysis identified six clusters of cells with memory characteristics displaying high expression of CD38, CD80 and/or CD273, and variable expression of IgD, all of which expressed high levels of FCRL5 (*Figure 6C*: clusters identified with purple arrows). We then used Isomap, (Cytofkit package) to infer the relatedness between those cell subsets identified by PhenoGraph. This confirmed high similarities between the cell clusters expressing high levels of FCRL5 with the clusters expressing high levels of the memory markers CD80, CD273 and CD38 (*Figure 6D*).

To confirm the memory identity of $MSP1_{21}$-specific $FCRL5^{hi}$ B-cells detected after resolution of the infection, we isolated $MSP1_{21}$-specific B cells expressing either high levels of FCRL5 or not expressing FCRL5 (i.e. $FCRL5^{hi}$ and $FCRL5^-$ $MSP1_{21}$-specific B cells) from the spleen of *P. chabaudi*-infected NIMP23→$Rag2^{-/-}$ mice (155dpi) (*Figure 6—figure supplement 1*), and $MSP1_{21}$-specific B

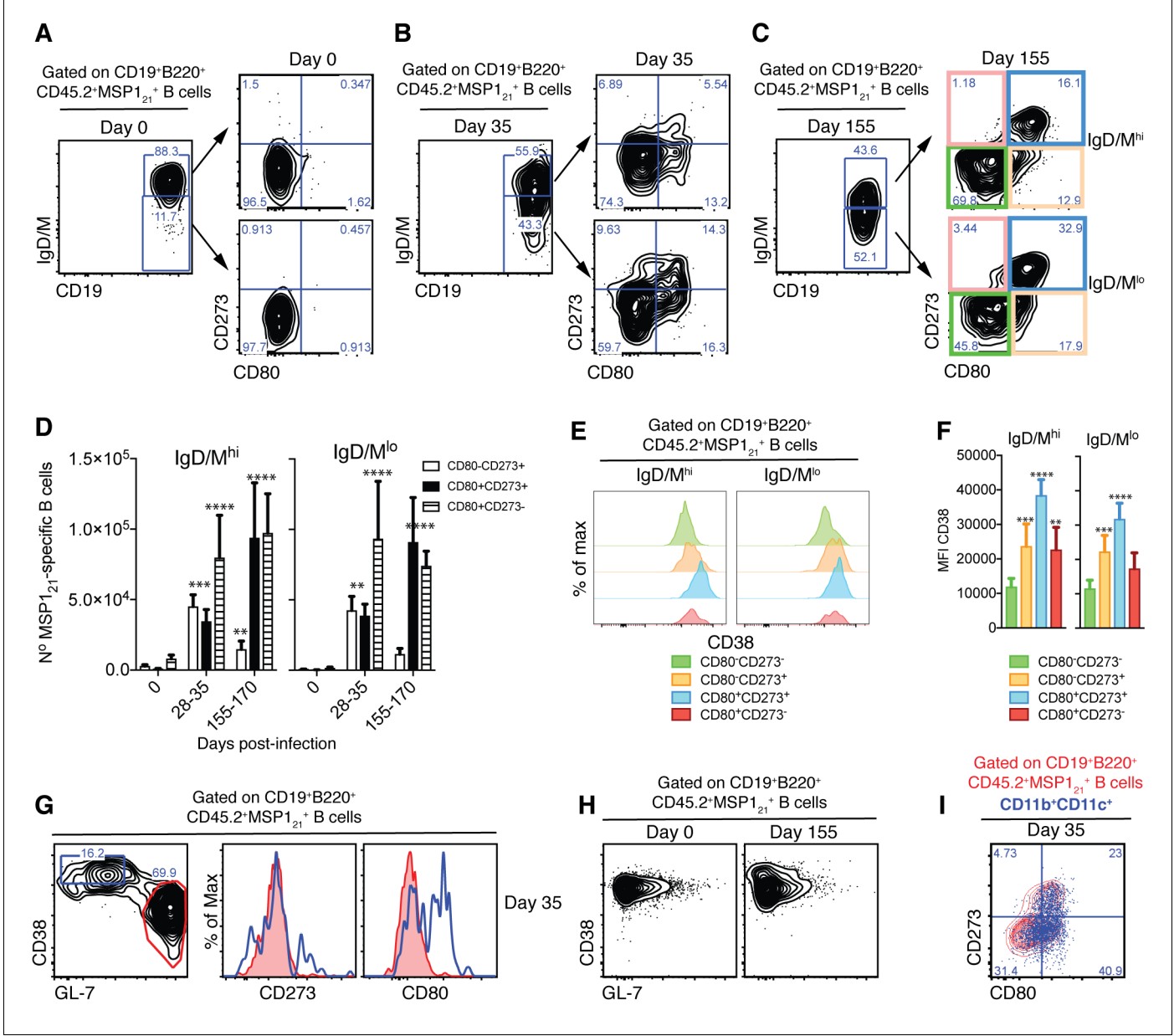

**Figure 5.** Detection of MSP1$_{21}$-specific B$_{mem}$ after resolution of *P.chabaudi* infection. (**A**), (**B**) and (**C**) Flow cytometry showing gating strategy to identify splenic IgM/D$^{hi}$ and IgM/D$^{lo}$ CD273$^+$ and/or CD80$^+$ MSP1$_{21}$-specific B$_{mem}$ in NIMP23→*Rag2*$^{-/-}$ chimeric mice before infection (day 0), at 35 and 155dpi, respectively. (**D**) Numbers of splenic IgM/D$^{hi}$ and IgM/D$^{lo}$ CD273$^+$ and/or CD80$^+$ MSP1$_{21}$-specific B$_{mem}$ in NIMP23→*Rag2*$^{-/-}$ chimeric mice during the course of mosquito transmitted *P. chabaudi* infection. Two-way ANOVA vs day 0. **p<0.01; ***p<0.001; ****p<0.0001. (**E**) Flow cytometry showing expression of CD38 on different subsets of splenic IgM/D$^{hi}$ and IgM/D$^{lo}$ MSP1$_{21}$-specific B cells from NIMP23→*Rag2*$^{-/-}$ chimeric mice defined based on CD273 and CD80 expression at 155dpi. (**F**) Geometric MFI of CD38 expression on different subsets of IgM/D$^{hi}$ and IgM/D$^{lo}$ splenic MSP1$_{21}$-specific B cells from NIMP23→*Rag2*$^{-/-}$ chimeric mice defined based on CD273 and CD80 expression at day 155 post-mosquito transmitted *P. chabaudi* infection. Two-way ANOVA vs CD273$^-$CD80$^-$ subset. **p<0.01; ***p<0.001; ****p<0.0001. Error bars are SEM. (**G**) Flow cytometry of CD273 and CD80 expression on non-GC (CD38$^{hi}$GL-7$^{lo}$, blue) and GC (CD38$^{lo}$GL-7$^{hi}$, red) splenic MSP1$_{21}$-specific B cells from NIMP23→*Rag2*$^{-/-}$ chimeric mice at 35dpi. (**H**) Flow cytometry of CD38 vs GL-7 (GC markers) on splenic MSP1$_{21}$-specific B cells from NIMP23→*Rag2*$^{-/-}$ chimeric mice at 0 and 155dpi. (**I**) CD11b$^+$CD11c$^+$ MSP1$_{21}$-specific B cells overlaid on the CD80 vs CD273 plot corresponding to total MSP1$_{21}$-specific B cells. Data pooled from three independent experiments with 3–7 mice per group.

DOI: https://doi.org/10.7554/eLife.39800.010

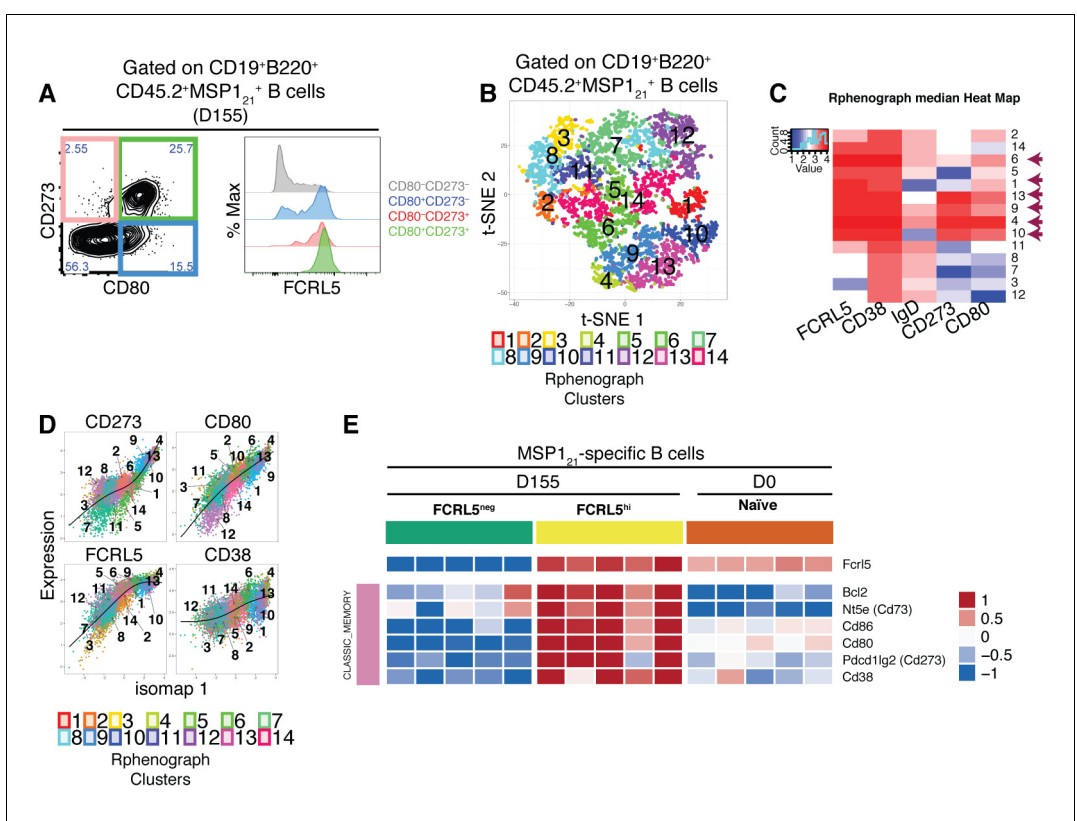

**Figure 6.** FCRL5^hi identifies MSP1$_{21}$-specific B$_{mem}$ after resolution of *P.chabaudi* infection. (**A**) Flow cytometry showing expression of FCRL5 (right) on different subsets of splenic MSP1$_{21}$-specific B cells from NIMP23→*Rag2*$^{-/-}$ chimeric mice defined based on CD273 and CD80 expression (left) at 155dpi. (**B**) t-SNE analysis of splenic MSP1$_{21}$-specific B cells based on FCRL5, CD38, IgD, CD273 and CD80 expression measured by flow cytometry (n = 5). Clusters identified by PhenoGraph are colored and numbered. (**C**) PhenoGraph heat map showing median expression of FCRL5, CD38, IgD, CD273 and CD80 on the different clusters of MSP1$_{21}$-specific B cells. Arrows point at the different clusters displaying a memory B cell phenotype. (**D**) Expression profiles of FCRL5, CD38, CD273 and CD80 for the different PhenoGraph clusters visualized on the first component of ISOMAP. The regression line estimated using the generalized linear model (GLM) is added for each marker. Data representative of three independent experiments with 4–7 mice per group. (**E**) Heat map showing expression levels of different genes on splenic FCRL5— and FCRL5^hi MSP1$_{21}$-specific B cells sorted at 155dpi, and MSP1$_{21}$-specific B cells sorted before infection (naïve), determined by RNAseq analysis. Each column corresponds to data from an individual mouse (n = 5 155 dpi, n = 5 0 dpi).

DOI: https://doi.org/10.7554/eLife.39800.011

The following figure supplements are available for figure 6:

**Figure supplement 1.** Gating strategy for the sorting of splenic MSP1$_{21}$-specific FCRL5^hi B$_{mem}$ and FCRL5— B cells at 155dpi.

DOI: https://doi.org/10.7554/eLife.39800.012

**Figure supplement 2.** High expression of FCRL5 identifies B$_{mem}$.

DOI: https://doi.org/10.7554/eLife.39800.013

cells from the spleen of naïve NIMP23→*Rag2*$^{-/-}$ mice, by flow cytometric sorting, and performed mRNAseq analysis on these three sorted cell populations (*Figure 6E*). As expected, the MSP1$_{21}$-specific FCRL5^hi B cell subset showed high expression of genes encoding the hallmark memory B cell markers *Cd38*, *Cd80*, *Cd86*, *Nt5e* (CD73) and *Pdcd1lg2* (CD273), when compared with either MSP1$_{21}$-specific FCRL5— B cells sorted at the same time or MSP1$_{21}$-specific B cells sorted from naïve mice (*Figure 6E*). Moreover, the MSP1$_{21}$-specific FCRL5^hi B cells sorted after resolution of the infection upregulated the anti-apoptotic *Bcl2* gene, which is an additional hallmark characteristic of memory B cells (*Figure 6E*). Importantly, FCRL5 also identified CD80$^+$ and CD273$^+$ MSP1$_{21}$-specific B$_{mem}$ subsets generated following immunization with a model antigen (*Figure 6—figure supplement 2*).

Thus, after resolution of the infection, high expression of FCRL5 identifies *P. chabaudi*-specific $B_{mem}$.

## *Plasmodium*-specific AMB are a distinct short-lived activated B cell subset

After identifying and sorting $MSP1_{21}$-specific AMB during chronic *P. chabaudi* infection, and $MSP1_{21}$-specific $B_{mem}$ after resolution of the infection, we then compared the transcriptome of these two B-cell subsets. Principal component analysis (PCA) demonstrated a strikingly distinct transcriptome of $MSP1_{21}$-specific AMB from that of $MSP1_{21}$-specific $B_{mem}$, as well as all other $MSP1_{21}$-specific B-cell subsets sorted in this study (*Figure 7A*). The $MSP1_{21}$-specific AMB sorted at 35dpi formed a separated cluster at the extreme right of the PC1 axis of the PCA plot, which accounts for the majority of the variance (*Figure 7A*). All the other subsets [including $MSP1_{21}$-specific $CD11b^-CD11c^-$ B-cells sorted from the same mice and at the same day post-infection as the $MSP1_{21}$-specific AMB (i.e. 35dpi)] clustered on the left of the PC1 axis, and showed differences mostly along the PC2 axis of the PCA plot, which accounts for only 10% of the variance (*Figure 7A*). Interestingly, $MSP1_{21}$-specific $CD11b^-CD11c^-$ and $MSP1_{21}$-specific $B_{mem}$ clustered on opposite sides of the $MSP1_{21}$-specific naïve B-cell subset along the PC2 axis (*Figure 7A*), which suggests that the $MSP1_{21}$-specific $B_{mem}$ more closely resemble $MSP1_{21}$-specific naïve B cells than $MSP1_{21}$-specific $CD11b^-CD11c^-$ B cells sorted at 35dpi.

$MSP1_{21}$-specific AMB sorted during chronic *P. chabaudi* infection, and $MSP1_{21}$-specific $B_{mem}$ sorted after resolution of the infection shared the expression of a series of mouse memory markers, including *Cd80*, *Fcrl5*, *Nt5e* (CD73), and *Cd86* (*Figure 7B*). However, these two subsets showed differences in the expression pattern of anti- and pro-apoptotic genes (*Figure 7C*). While $MSP1_{21}$-specific $B_{mem}$ from after infection resolution showed the highest levels of expression of the anti-apoptotic *Bcl2* gene, $MSP1_{21}$-specific AMB sorted during chronic *P. chabaudi* infection showed the lowest levels of expression of this hallmark anti-apoptotic gene (*Figure 7C*). In contrast to $MSP1_{21}$-specific $B_{mem}$, $MSP1_{21}$-specific AMB expressed high levels of the pro-apoptotic genes *Bad*, *Bax*, *Fas* and *Fasl* (*Figure 7C*). Interestingly, $MSP1_{21}$-specific AMB expressed very high levels of class-switched immunoglobulins, including *Igha*, *Ighg1*, *Ighg2b*, *Ighg2c* and *Ighg3* (*Figure 7D*). Finally, $MSP1_{21}$-specific AMB highly expressed *Cd11b*, *Cd11c*, *Tbx21*, *Ifng* and *Pdcd1* (*Figure 7E*), all hallmarks of human AMB, as well as *Mki67*, indicative of active cell division, as previously shown in human AMB.

These data demonstrate that AMB and $B_{mem}$ share the expression of memory markers. However, they show striking differences in the expression of pro- and anti-apoptotic genes, immunoglobulins genes, and cell proliferation genes. The increased expression of *Mki67*, pro-apoptotic genes and class-switched immunoglobulins in AMB suggests that they resemble activated B cells. By contrast, $B_{mem}$ express much less *Mki67* (similar to naïve cells) and present an anti-apoptotic gene expression pattern, consistent with being long-lived quiescent B cells.

Mouse FCRL5 has been shown to be expressed on marginal zone (MZ) and B1 B cells (*Won et al., 2006*). Therefore, it is possible that the AMB identified in this study might represent a specific subset of either MZ or B1 B cells. As discussed elsewhere (*Baumgarth, 2011*; *Garraud et al., 2012*; *Pillai et al., 2005*; *Zouali, 2011*), B1 and MZ B cells are $CD1d^{mid/hi}$, $CD9^+$, $IgM^{hi}$ and $CD23^-$. B1 are further $CD43^+$, $B220^{lo}$, and may (B1a) or may not (B1b) express CD5. MZ are further characterised by $CD22^{hi}$, $CD21/CR2^{hi}$, and the expression of the lysophospholipid sphingosine-1 phosphate receptor $S1P_1$, and the lineage master regulator Notch2. Indeed, $MSP1_{21}$-specific $CD11b^+CD11c^+$ AMB showed high expression of some markers associated with B1 and/or MZ, including CD5, CD9 and CD43. However, these markers have also been shown to be highly expressed by activated B2 B cells and plasma cells (*Baumgarth, 2011*). Studying all these canonical markers, we found no other similarities between B1/MZ and $MSP1_{21}$-specific $CD11b^+CD11c^+$ AMB to suggest a relationship between these subsets (*Figure 8A*). This was further corroborated by flow cytometry analysis (*Figure 8B*). Moreover, unlike MZ and B1 B cells, $MSP1_{21}$-specific AMB were class-switched and expressed very high levels of IgG (*Figure 7D*). To further characterize the identity of $MSP1_{21}$-specific $CD11b^+CD11c^+$ AMB, we explored if these cells resemble GC B cells. Flow cytometry analysis demonstrated that $MSP1_{21}$-specific AMB did not display a GC phenotype (i.e. $CD38^{hi}GL-7^{lo}$) (*Figure 8C*) while the $MSP1_{21}$-specific $CD38^{lo}GL-7^{hi}$ GC B cells showed low expression of both CD11b and CD11c. Thus, $MSP1_{21}$-specific AMB are not GC B cells.

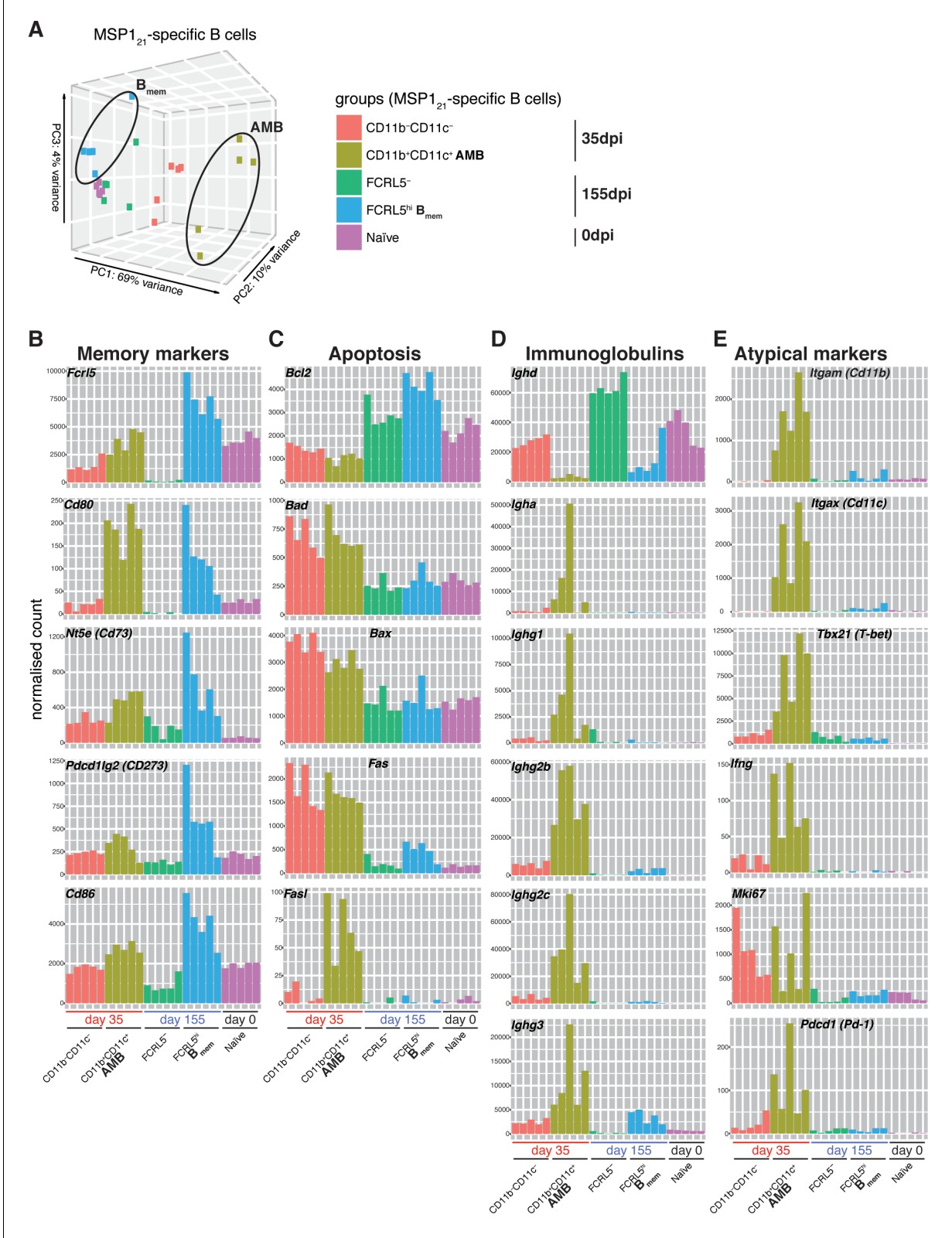

**Figure 7.** MSP1₂₁-specific AMB are a distinct short-lived activated B cell subset. (**A**) Principal component analysis of RNAseq transcriptome data from splenic MSP1₂₁-specific AMB (CD11b⁺CD11c⁺, 35dpi), CD11b⁻CD11c⁻ B cells (35dpi), B_mem (FCRL5ʰⁱ, 155dpi), FCRL5⁻ B cells (155dpi) and B cells from naïve mice (0dpi). The MSP1₂₁-specific AMB and B_mem are contained inside ellipses. (**B**) (**C**) (**D**) (**E**) Normalized counts corresponding to selected
*Figure 7 continued on next page*

*Figure 7 continued*

genes representing memory B-cell markers, anti and pro-apoptotic genes, immunoglobulins and atypical memory B-cell markers, respectively, for all five groups described in (A). Each bar represents an individual mouse. Data generated with five mice per group.

DOI: https://doi.org/10.7554/eLife.39800.014

All together, these data further show that *P. chabaudi*-specific AMB represent a distinct subset of short-lived activated B cells.

## Discussion

Similar to other chronic infections [e.g. HIV, HCV and *Mycobacterium tuberculosis* (*Knox et al., 2017b*; *Portugal et al., 2017*)], *Plasmodium* infection, the cause of malaria, leads to an increase in the frequency of AMB [originally termed tissue-like memory B cells (*Moir et al., 2008*)] in peripheral blood from *P. falciparum*-exposed subjects (*Illingworth et al., 2013*; *Portugal et al., 2015*; *Sullivan et al., 2016*; *Sullivan et al., 2015*; *Weiss et al., 2011*; *Weiss et al., 2010*; *Weiss et al., 2009*). However, mouse models to investigate *Plasmodium*-specific AMB are lacking. Here we have generated an IgH knock-in transgenic mouse strain to study the generation of *Plasmodium*-specific AMB in a *Plasmodium chabaudi* infection. We demonstrate the generation of *P. chabaudi*-specific AMB in response to blood-stage mosquito-transmitted chronic *P. chabaudi* infection, and show the short-lived nature of these cells. *P. chabaudi* infections in mice present some outstanding characteristics for the study of AMB; a chronic phase which allows investigation of the impact of constant immune activation driven by persistent subpatent parasitemia, followed by a clearance phase which allows the study of immune responses after the infection is naturally resolved. Thus, it is possible to study both exhaustion driven by chronic immune activation, and memory immune responses which remain after *P. chabaudi* elimination.

The *P. chabaudi*-specific AMB we detected during the chronic phase of infection showed strong similarities to human AMB described in chronic malaria. These included being class-switched, and expressing mouse homologues of hallmark human atypical memory B-cell genes such as *Itgax* (Cd11b), *Itgam* (Cd11c), *Cxcr3*, *Fcrl5*, *Tbx21* (T-bet), *Ifng*, *Cd80*, *Cd86*, *Aicda*, and a series on inhibitory receptors, including *Lair1* and *Pdcd1* (PD-1). Human AMB have been shown to share several characteristics with plasmablasts (*Knox et al., 2017a*; *Muellenbeck et al., 2013*; *Obeng-Adjei et al., 2017*; *Sullivan et al., 2015*). Interestingly, we also observed increased expression of several genes associated with plasmablasts in *P. chabaudi*-specific AMB, including *Cd138*, *Xbp1*, *Prdm1* (encoding Blimp-1) and *Mki67*, accompanied by downregulation of *Pax5* and *Bcl6*.

AMB resemble age-associated B cells (ABC) which accumulate with age as well as in autoimmunity, and were also proposed to be a subset of long-lived memory B cells (*Naradikian et al., 2016a*; *Portugal et al., 2017*; *Rubtsov et al., 2017*). Expression of T-bet, CD11c and CXCR3 are shared by AMB, tissue-like memory B cells, and ABC (*Knox et al., 2017b*; *Naradikian et al., 2016a*). Moreover, similar to ABC, expansion of human AMB associated with malaria is driven by IFNγ (*Obeng-Adjei et al., 2017*). IL-21, which is highly expressed by follicular helper T cells in response to *Plasmodium* infection (*Carpio et al., 2015*; *Obeng-Adjei et al., 2015*; *Pérez-Mazliah et al., 2015*), also directly promotes T-bet expression in B cells in the context of TLR engagement (*Naradikian et al., 2016b*). Taken together, these data strongly suggest that this is the same T-bet$^+$ B-cell subset, which accumulates with time due to repetitive antigenic exposure. In agreement with previous data (*Rubtsova et al., 2013*), we show here that immunization with MSP1$_{21}$ and R848, a TLR7/8 ligand, promotes a robust but short-lived CD11b$^+$CD11c$^+$ *P. chabaudi*-specific AMB response. T-bet$^+$ atypical B cells are critical to eradicate various murine viral infections (*Barnett et al., 2016*; *Rubtsova et al., 2013*), and a recent study showed that yellow fever and vaccinia vaccinations of humans stimulated an acute T-bet$^+$ B-cell response and suggested that these T-bet$^+$ B-cell population may function as an early responder during acute viral infections (*Knox et al., 2017a*). Thus, T-bet$^+$ B cells, even in the context of malaria, are likely to be a normal component of the immune compartment that becomes activated and expands, most probably in response to BCR, endosomal TLR, and IFNγ or IL-21 stimulation. Moreover, a recent study shows a TLR9/IFNγ-dependent activation of autoreactive T-bet$^+$CD11c$^+$ atypical B cells in response to *P. yoelii* 17XNL infection in mice

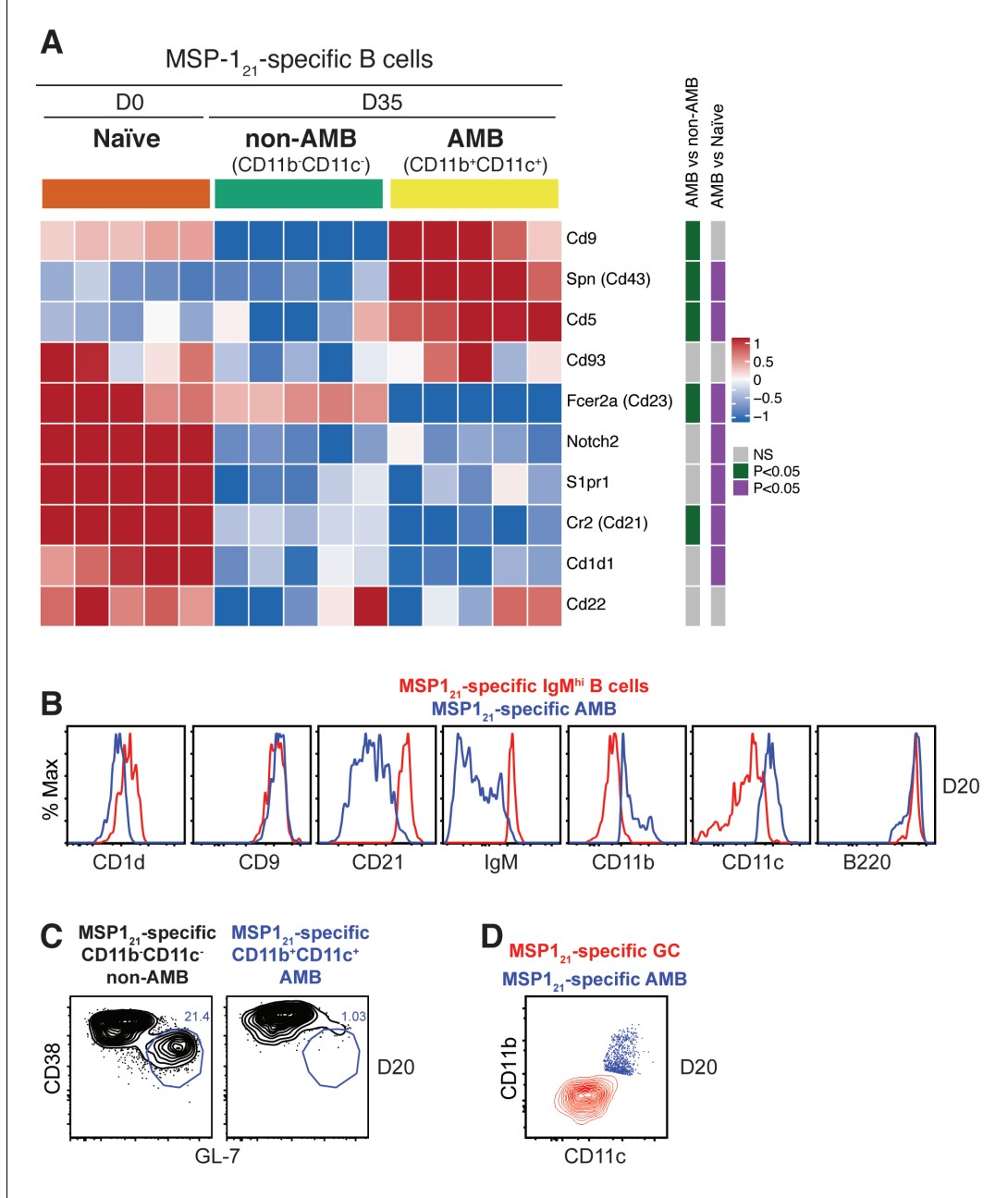

**Figure 8.** Analysis of MZ, B1 and GC B cell characteristics on MSP1$_{21}$-specific AMB. (A) MSP1$_{21}$-specific CD11b$^+$CD11c$^+$ (AMB) and CD11b$^-$CD11c$^-$ B cells were flow cytometry sorted from the spleen of NIMP23→$Rag2^{-/-}$ chimeric mice at 35dpi; MSP1$_{21}$-specific B cells were flow cytometry sorted from the spleen of naïve NIMP23→$Rag2^{-/-}$, and these three B cell populations were submitted to mRNAseq analysis. The heat map displays level of expression of selected individual genes known to be up or downregulated on either MZ, B1 B cells or both. Each column represents an individual mouse. (B) Flow cytometry analysis of surface markers of either MZ, B1 B cells or both, on MSP1$_{21}$-specific CD11b$^+$CD11c$^+$ (AMB) (blue) and MSP1$_{21}$-specific IgM$^{hi}$ (red) B cells from the spleen of $Igh^{NIMP23/+}$ mice at 20dpi. (C) Flow cytometry analysis of GC markers on MSP1$_{21}$-specific CD11b$^-$CD11c$^-$ (non-AMB, left) and CD11b$^+$CD11c$^+$ (AMB, right) B cells from the spleen of $Igh^{NIMP23/+}$ mice at 20dpi. (D) Flow cytometry analysis showing the expression of CD11b and CD11c on MSP1$_{21}$-specific CD11b$^+$CD11c$^+$ (AMB, blue) compared to GC (CD38$^{lo}$GL-7$^{hi}$, red) B cells from the spleen of $Igh^{NIMP23/+}$ mice. Data generated with 5–6 mice per group.

DOI: https://doi.org/10.7554/eLife.39800.015

(*Rivera-Correa et al., 2017*). However, whether these cells remained part of the long-lived memory B-cell pool after resolution of the infection was not explored.

Here, we show that *P. chabaudi*-specific AMB are short-lived activated B cells. These cells were absent after resolution of the infection, and immunization with purified antigen and TLR agonists resulted in a transient, yet robust, activation of *P. chabaudi*-specific AMB which lasted no more than 48 hr. Moreover, the $Igh^{NIMP23/+}$ mouse model allowed us to obtain a deep insight of the transcriptome profile of MSP1$_{21}$-specific AMB during natural infection, and compare it side-by-side with the transcriptome of MSP1$_{21}$-specific naïve B cells. This allowed us to demonstrate their heavily pro-apoptotic and activated transcription profile, further explaining their short-lived nature. *P. chabaudi*-specific AMB showed very low expression of *Bcl2*, and high levels of expression of several pro-apoptotic genes including *Bad*, *Bax*, *Fas* and *Fasl*. In addition, these cells expressed very high levels of class-switched immunoglobulins and genes associated with DNA replication and proliferation.

The association of *P. chabaudi*-specific AMB with ongoing infection explains several observations in human studies: reduction of HIV plasma viremia by ART resulted in a significant reduction of HIV-specific AMB without altering the frequency of HIV-specific B$_{mem}$ (*de Bree et al., 2017*; *Kardava et al., 2014*); individuals living in high malaria endemicity present higher frequencies of AMB than individuals living in areas with moderate transmission (*Illingworth et al., 2013*; *Sullivan et al., 2015*); repetitive *Plasmodium* episodes result in higher frequencies of AMB (*Obeng-Adjei et al., 2017*); the percentage of AMB is larger in children with persistent asymptomatic *Plasmodium falciparum* parasitemia as compared with parasite-free children (*Weiss et al., 2009*); previously exposed subjects significantly reduce the frequency of AMB following a year of continuous absence of exposure to *Plasmodium falciparum* infection (*Ayieko et al., 2013*). These observations all support the view that constant immune activation rather than impaired memory function leads to the accumulation of AMB in malaria.

After resolution of infection, *P. chabaudi*-specific AMB did not persist, but instead subsets of *P. chabaudi*-specific B$_{mem}$ were readily detected. These cells expressed different combinations of previously described mouse B-cell memory markers [i.e. CD80, CD273 and CD73 (*Anderson et al., 2007*; *Tomayko et al., 2010*; *Zuccarino-Catania et al., 2014*)]. *P. chabaudi*-specific B$_{mem}$ included both class-switched and non-class-switched cells, which show different responses to a secondary challenge infection (*Krishnamurty et al., 2016*). Independently of the combination of previously described memory markers expressed on *P. chabaudi*-specific B$_{mem}$, all of these cells displayed very high expression of FCRL5. Previous data showed most prominent expression of FCRL5 in marginal zone B cells, while much less evident in the newly-formed and follicular splenic B cell subpopulations (*Davis et al., 2004*; *Won et al., 2006*). In agreement with this, we observed expression of FCRL5 on a subset of splenic *P. chabaudi*-specific B cells obtained from naïve mice. However, the level of FCRL5 expression on *P. chabaudi*-specific B$_{mem}$ detected after resolution of the infection was noticeably higher than that of naïve B cells both at the protein and mRNA levels. In contrast to MSP1$_{21}$-specific AMB, these CD11b⁻CD11c⁻FCRL5$^{hi}$ MSP1$_{21}$-specific B$_{mem}$ showed high expression of the hallmark memory and anti-apoptotic gene *Bcl2* (*Bhattacharya et al., 2007*). Thus, after resolution of the infection, high expression of FCRL5 acted as a universal B$_{mem}$ marker. Due to its complex dual ITIM/ITAM signaling capacity (*Zhu et al., 2013*), it is tempting to speculate that FCRL5 might serve as an important signal in the differentiation/maintenance of B$_{mem}$.

Tracking the fate of the different MSP1$_{21}$-specific B-cell subsets identified in this work will allow detailing the interplay between them. A model can be proposed in which antigen-specific AMB serve as an intermediate stage of differentiation between naïve and B$_{mem}$. Alternatively, antigen-specific AMB might represent early plasmablasts or recent GC emigrates, based on class-switching and the high expression of IgG by these cells. These scenarios are not necessarily antagonist, and might even occur in parallel. Moreover, B1 B cells have been shown to class-switch and contribute to serum IgG1, IgG2a and IgA to influenza (*Baumgarth et al., 2005*), and IgG-producing B1a B cells have been shown to accumulate in the spleen of a mouse model of systemic lupus erythematosus (*Enghard et al., 2010*). Therefore, we can't rule out the possibility that AMB might represent a particular B1 B cell subset that expands in the spleen and blood in response to *Plasmodium* infection.

Our data suggest that the expansion of AMB in malaria is not a consequence of B-cell exhaustion, but rather a physiologic stage of B-cell activation, and that these cells are sustained in high frequencies by ongoing chronic infections. Thus, *Plasmodium*-specific AMB are neither 'memory', nor 'atypical'. Importantly, our data demonstrate that robust expansion of *Plasmodium*-specific AMB does not

hinder clearance of the infection, activation of germinal centers, or generation of *Plasmodium*-specific long-lived quiescent $B_{mem}$ upon resolution of the infection.

# Materials and methods

**Key resources table**

| Reagent type (species) or resource | Designation | Source or reference | Identifiers | Additional information |
|---|---|---|---|---|
| Genetic reagent (*M. musculus*) | B6.SJL-Ptprc[a] Pepc[b]/BoyJ (B6.CD45.1) | The Jackson Laboratory | MGI:4819849 | Bred in the specific pathogen-free facilities of the MRC National Institute for Medical Research and The Francis Crick Institute |
| Genetic reagent (*M. musculus*) | Rag2[tm1Fwa] (*Rag2[-/-]*) | The Jackson Laboratory | MGI:1858556 | Bred in the specific pathogen-free facilities of the MRC National Institute for Medical Research and The Francis Crick Institute |
| Genetic reagent (*M. musculus*) | *Igh*[NIMP23/+] | This paper | _ | Bred in the specific pathogen-free facilities of the MRC National Institute for Medical Research and The Francis Crick Institute |
| Strain, strain background (*Plasmodium chabaudi chabaudi*, strain AS) | *P. chabaudi* | other | _ | European Malaria Reagent Repository, University of Edinburgh. |
| Strain, strain background (*Anopheles stephensi*, strain SD500, female) | mosquitos | PMID: 23217144 | _ | Bred in Jean Langhorne's lab |
| Antibody | Monoclonal Rat Anti-CD11b | BD Biosciences | 563553 | (dil 1/50) |
| Antibody | Monoclonal Hamster Anti-Mouse CD11c | BD Biosciences | 561022 | (dil 1/50) |
| Antibody | Monoclonal Rat Anti-Mouse CD138 | BD Biosciences | 553714 | (dil 1/400) |
| Antibody | Monoclonal Rat Anti-Mouse CD19 | BD Biosciences | 565076 | (dil 1/200) |
| Antibody | Monoclonal Rat anti-Mouse CD19 | Biolegend | 115530 | (dil 1/400) |
| Antibody | Monoclonal Rat anti-Mouse CD19 | Biolegend | 115543 | (dil 1/400) |

*Continued on next page*

*Continued*

| Reagent type (species) or resource | Designation | Source or reference | Identifiers | Additional information |
|---|---|---|---|---|
| Antibody | Monoclonal Rat anti-Mouse CD1d | Biolegend | 123510 | (dil 1/100) |
| Antibody | Monoclonal Rat anti-mouse CD2 | Biolegend | 100112 | (dil 1/100) |
| Antibody | Monoclona Rat Anti-Mouse CD21/35 | BD Biosciences | 563176 | (dil 1/100) |
| Antibody | Monoclona Rat Anti-Mouse CD21/35 | BD Biosciences | 553818 | (dil 1/100) |
| Antibody | Monoclonal Rat anti-Mouse CD23 | eBioscience | 25–0232 | (dil 1/100) |
| Antibody | Monoclonal Rat anti-Mouse CD273 | BD Biosciences | 564245 | (dil 1/25) |
| Antibody | Armenian Hamster anti-Mouse CD3 | Biolegend | 100336 | (dil 1/100) |
| Antibody | Monoclonal Rat anti-Mouse CD38 | Biolegend | 102718 | (dil 1/400) |
| Antibody | Monoclonal Rat anti-Mouse CD38 | eBioscience | 17–0381 | (dil 1/400) |
| Antibody | Monoclonal Rat anti-Mouse CD38 | BD Biosciences | 740697 | (dil 1/400) |
| Antibody | Monoclonal Rat anti-Mouse CD4 | Biolegend | 100414 | (dil 1/400) |
| Antibody | Monoclonal Rat anti-Mouse CD45.1 | Biolegend | 110706 | (dil 1/400) |
| Antibody | Monoclonal Rat anti-Mouse CD45.1 | Biolegend | 110728 | (dil 1/400) |
| Antibody | Monoclonal Mouse anti-Mouse CD45.2 | BD Biosciences | 563685 | (dil 1/50) |
| Antibody | Monoclonal Mouse anti-Mouse CD45.2 | Biolegend | 109814 | (dil 1/50) |
| Antibody | Monoclonal Mouse anti-Mouse CD45.2 | Biolegend | 109808 | (dil 1/50) |
| Antibody | Monoclonal Rat anti-Mouse CD45R/B220 | BD Biosciences | 564449 | (dil 1/400) |
| Antibody | Monoclonal Rat anti-Mouse CD45R/B220 | Biolegend | 103224 | (dil 1/400) |

*Continued on next page*

*Continued*

| Reagent type (species) or resource | Designation | Source or reference | Identifiers | Additional information |
|---|---|---|---|---|
| Antibody | Monoclonal Rat anti-Mouse CD45R/B220 | eBioscience | 25–0452 | (dil 1/400) |
| Antibody | Monoclonal Rat anti-Mouse CD73 | BD Biosciences | 550741 | (dil 1/100) |
| Antibody | Monoclonal Armenian Hamster anti-Mouse CD80 | Biolegend | 104729 | (dil 1/25) |
| Antibody | Monoclonal Rat anti-Mouse CD8a | Biolegend | 100734 | (dil 1/400) |
| Antibody | Monoclonal Rat anti-Mouse CD9 | BD Biosciences | 558749 | (dil 1/100) |
| Antibody | Monoclonal Rat anti-Mouse CD93 (AA4.1) | eBioscience | 17–5892 | (dil 1/100) |
| Antibody | FCRL5 | PMID: 17082595 | _ | Produced in Randall Davis' lab (dil 1/400) |
| Antibody | Polyclonal Sheep anti-Mouse FCRL5 | R and D Systems | FAB6756G | (dil 1/50) |
| Antibody | Monoclonal Rat Anti-Mouse T- and B-Cell Activation Antigen GL7 | BD Biosciences | 562080 | (dil 1/100) |
| Antibody | Monoclonal Rat Anti-Mouse IgD | Biolegend | 405725 | (dil 1/400) |
| Antibody | Monoclonal Rat Anti-Mouse IgD | Biolegend | 405723 | (dil 1/400) |
| Antibody | Monoclonal Rat Anti-Mouse IgD | Biolegend | 405710 | (dil 1/400) |
| Antibody | Monoclonal Rat Anti-Mouse IgG2b | Biolegend | 406708 | (dil 1/25) |
| Antibody | Monoclonal Rat Anti-Mouse IgM | Biolegend | 406512 | (dil 1/100) |
| Recombinant DNA reagent | $VDJ_H^{NIMP23}$ anti-$MSP1_{21}$ variable region coding exon containing the Leader-V segment intron from gDNA of the NIMP23 hybridoma | PMID: 7141700 | _ | Produced in Jean Langhorne's lab |

*Continued on next page*

*Continued*

| Reagent type (species) or resource | Designation | Source or reference | Identifiers | Additional information |
|---|---|---|---|---|
| Recombinant DNA reagent | C57Bl/6 IgH HEL variable region knock-in construct | PMID: 12668643 | _ | Donated by Robert Brink of the Garvan Institute of Medical Research, New South Wales, Australia |
| Peptide, recombinant protein | MSP1$_{21}$ | PMID: 11254580 | _ | Produced in Jean Langhorne's lab |
| Commercial assay or kit | EZ-Link Sulfo-NHS-LC-Biotinylation Kit | Thermo Scientific | 21435 | |
| Commercial assay or kit | RiboPure RNA Purification Kit | Invitrogen | AM1924 | |
| Commercial assay or kit | Qubit 1X dsDNA HS Assay Kit | Invitrogen | Q33231 | |
| Commercial assay or kit | SMART-Seq v4 Ultra Low Input RNA Kit for Sequencing | Takara | 634889 | |
| Commercial assay or kit | Ovation Ultralow Library System V2 | Nugen | 0344–32 | |
| Commercial assay or kit | LIVE/DEAD Fixable Aqua Dead Cell Stain Kit | Invitrogen | L34957 | |
| Commercial assay or kit | LIVE/DEAD Fixable Blue Dead Cell Stain Kit | Invitrogen | L23105 | |
| Chemical compound, drug | Streptavidin-R-Phycoerythrin | Prozyme | PJRS25 | |
| Chemical compound, drug | Streptavidin-Allophycocyanin | Prozyme | PJ27S | |
| Chemical compound, drug | TiterMax Gold Adjuvant | Merck (formerly Sigma-Aldrich) | T2684-1ML | |
| Chemical compound, drug | R848 (Resiquimod) | Invivogen | tlrl-r848 | |
| Chemical compound, drug | TRI Reagent Solution | Invitrogen | AM9738 | |
| Software, algorithm | cutadapt v1.9.1 | doi:10.14806/ej.17.1.200 | | |
| Software, algorithm | RSEM v1.2.31 | doi:10.1186/1471-2105-12-323 | | |
| Software, algorithm | STAR v2.5.1b | doi:10.1093/bioinformatics/bts635 | | |
| Software, algorithm | DESeq2 | doi:10.1186/s13059-014-0550-8 | | |
| Software, algorithm | R v3.4.0 | other | | https://www.r-project.org |
| Software, algorithm | Bioconductor v3.5 | other | | http://www.bioconductor.org |
| Software, algorithm | Broad's Gene Set Enrichment Analysis (GSEA) | other | | http://software.broadinstitute.org/gsea/index.jsp |

*Continued on next page*

*Continued*

| Reagent type (species) or resource | Designation | Source or reference | Identifiers | Additional information |
|---|---|---|---|---|
| Software, algorithm | FlowJo version 9.6 or higher | Tree Star | | |
| Software, algorithm | Cytofkit | doi:10.1371/journa l.pcbi.1005112.s009 | | |
| Software, algorithm | Prism v6 | GraphPad | | |

## Mice

5-12 week-old female mice were used for experiments. C57BL/6J, C57BL/6.SJL-*Ptprc*[a] (CD45.1 congenic), *Rag2*[-/-].C57BL/6.SJL-*Ptprc*[a] (CD45.1 congenic) and BALB/c mouse strains were bred in the specific pathogen-free facilities of the MRC National Institute for Medical Research and The Francis Crick Institute, and were housed conventionally with sterile bedding, food and irradiated water. Room temperature was 22°C with a 12 hr light/dark cycle; food and water were provided ad libitum. The study was carried out in accordance with the UK Animals (Scientific Procedures) Act 1986 (Home Office license 80/2538 and 70/8326), was approved by the MRC National Institute for Medical Research Ethical Committee and was approved by The Francis Crick Institute Ethical Committee.

To produce MSP1$_{21}$–specific B cell knock-in mice capable of undergoing class switch recombination on the C57BL/6J genetic background, the VDJ$_H$$^{NIMP23}$ anti-MSP1$_{21}$ variable region coding exon containing the Leader-V segment intron from gDNA of the NIMP23 hybridoma (*Boyle et al., 1982*) was inserted by homologous recombination into the 5' end of the endogenous IgH locus (*Taki et al., 1993*) (*Figure 1—figure supplement 1A–C*). The VDJ$_H$$^{NIMP23}$ anti-MSP1$_{21}$ variable region coding exon was inserted into a previously described IgH targeting construct, replacing the anti-HEL heavy chain variable region coding exon that was already in it (*Phan et al., 2003*) (*Figure 1—figure supplement 1D*). The final targeting construct included a loxP-flanked neomycin resistance cassette in reverse transcriptional orientation to the *Igh* locus, located immediately 5' to the rearranged VDJ$_H$$^{NIMP23}$ variable region and its associated promoter (*Figure 1—figure supplement 1D*). Electroporation of C57BL/6N-derived PRX embryonic stem cells with the targeting construct and selection of homologous recombinant clones was performed using standard techniques by Poly-Gene AG (Switzerland). One targeted ES clone was used for production of chimeric mice using standard techniques at the Biological Research Facilities of the MRC National Institute for Medical Research, London, UK. Male chimeric mice were crossed to C57BL/6J females and progeny carrying the *Igh*$^{NIMP23neo}$ allele were crossed to PC3Cre mice (*O'Gorman et al., 1997*) to delete the neo$^r$ gene in the germline and generate mice carrying the *Igh*$^{NIMP23}$ allele (*Figure 1—figure supplement 1D*). The *Igh*$^{NIMP23/+}$ strain was maintained by backcrossing for at least 10 generations to C57BL/6J mice.

## Mixed bone marrow chimeras

Femurs and tibias were excised from female mice and cleaned of flesh using forceps and scalpel, and BM was obtained by flushing out with IMDM supplemented with 2 mM L-glutamine, 0.5 mM sodium pyruvate, 100 U penicillin, 100 mg streptomycin, 6 mM Hepes buffer, and 50 mM 2-ME (Gibco, Invitrogen), using a syringe with a needle. Thereafter, single BM cell suspensions were obtained by mashing through a 70 μm filter mesh, further sieved through 40 μm filter mesh and washed once. Live cells were resuspended in sodium chloride solution 0.9% (Sigma) at $4 \times 10^6$ cells/200 μl. *Rag2*$^{-/-}$.C57BL/6.SJL-*Ptprc*[a] mice were sub-lethally irradiated (5Gy) using a [137Cs] source and reconstituted less than 24 hr after irradiation by i.v. injections of a 10% *Igh*$^{NIMP23/+}$:90% C57BL/6.SJL-*Ptprc*[a] combination of donor BM cells. Recipient mice were maintained on acidified drinking water and analyzed for reconstitution after 6–8 weeks.

## *Plasmodium chabaudi* infection

*Plasmodium chabaudi chabaudi AS* was transmitted by *Anopheles stephensi* mosquitoes, strain SD500, as described elsewhere (*Spence et al., 2012*). Briefly, C57BL/6J mice were injected i.p. with $10^5$ *P. chabaudi*-infected red blood cells and used to feed mosquitos two weeks after the injection.

Two weeks after mosquito feeding/infection, each experimental mouse was exposed to 20 infected mosquitos for 30 min. Blood parasitemia in infected experimental mice was routinely monitored by thin blood smears.

## Immunizations

Mice were immunized i.p. with a combination of 100 µg of $MSP1_{21}$ (*Quin and Langhorne, 2001*) and 50 µl of Titermax Gold emulsion (Sigma), or a combination of 50 µg of $MSP1_{21}$ and 50 µg of R848 (Invivogen).

## Flow cytometry and cell sorting

Spleens, lymph nodes and bone marrows were dissected and single cell suspensions were obtained by mashing the organs through a 70 µm filter mesh in HBSS, 6 mM Hepes buffer (Gibco, Invitrogen). After removal of red blood cells from spleens and bone marrows by treatment with lysing buffer (Sigma), the remaining cells were resuspended in complete Iscove's Modified Dulbecco's Medium [IMDM supplemented with 10% FBS Serum Gold (PAA Laboratories, GE Healthcare), 2 mM L-glutamine, 0.5 mM sodium pyruvate, 100U penicillin, 100 mg streptomycin, 6 mM Hepes buffer, and 50 mM 2-ME (all from Gibco, Invitrogen)] and viable cells were counted using trypan blue (Sigma) exclusion and a hemocytometer. Cells were then resuspended in PBS and incubated with APC- or PE-labelled $MSP1_{21}$ fluorescent probes and/or different combinations of fluorochrome-conjugated antibodies (key resources table), and either acquired after two washes with PBS, or fixed with 2% paraformaldehyde and stored in staining buffer at 4°C until acquisition.

The APC and PE $MSP1_{21}$ fluorescent probes were produced as previously described (*Krishnamurty et al., 2016*; *Taylor et al., 2012*). Briefly, purified $MSP1_{21}$ (*Quin and Langhorne, 2001*) was biotinylated using an EZ-link Sulfo-NHS-LC- Biotinylation kit (Thermo Fisher Scientic) using a 1:1 ratio of biotin to protein, and loaded onto Streptavidin-APC conjugated or Phycolink Streptavidin-R-PE conjugated (ProZyme) in a 6:1 ratio of $MSP1_{21}$:Streptavidin-fluorochrome.

Cell sorting was performed on a MoFlo XDP (Beckman Coulter) or a BD FACSAria Fusion (BD Biosciences) and the target cell populations were directly dispensed into TRIreagent (Ambion) and stored at −80°C until RNA isolation. Purity checks were routinely performed for all assays by sorting aliquots of cells into PBS containing 2% FCS and reacquiring them on the cell sorter.

Dead cells were routinely excluded from the analysis by staining with LIVE/DEAD Fixable Aqua or Blue stain (Invitrogen). Singlets were selected based on FCS-A vs FCS-H and further based on SSC-A vs SSC-H. 'Fluorescence minus one' (FMO) controls were routinely used to verify correct compensation and to set the thresholds for positive/negative events. Analysis was performed with FlowJo software version 9.6 or higher (Tree Star).

PhenoGraph and *t*-distributed stochastic neighbor embedding (t-SNE) were combined to analyze multiparameter flow cytometry data using the Cytofkit package (*Chen et al., 2016*). t-SNE renders high-dimensional single-cell data based on similarities into only two dimensions, and thus helps visualize multiparameter data (*van der Maaten, 2008*). PhenoGraph (*Levine et al., 2015*) allows partitioning of high-dimensional single-cell data into phenotypically coherent subpopulations (i.e. clusters). The relatedness of the cell clusters identified by PhenoGraph was inferred using Isomap (Cytofkit package), in which related clusters/subsets can be visualised close to each other.

## RNA isolation, sequencing and data analysis

Total RNA from $1-5 \times 10^4$ cells sorted into TRIreagent (Ambion) was isolated using the Ribopure kit (Ambion). Concentration of purified RNA was determined by Qubit fluorometric quantitation using the HS assay kit (ThermoFisher Scientific), and the quality analyzed with a 2100 Bioanalyzer (Agilent). Samples with a RIN score above 8.50 were used for the next steps. cDNA was generated from total RNA with the SMART-Seq v4 Ultra Low Input RNA Kit (Takara Bio USA). Next-generation sequencing libraries were produced with the Ovation Ultralow System V2 (Nugen), and run as PE100 on a HiSeq 4000 sequencer (Illumina). GEO accession: GSE115155.

For bioinformatics analysis, paired-end sequence reads were adapter and quality trimmed using cutadapt v1.9.1 (*Martin, 2011*) with the following non-default settings: '-a AGATCGGAAGAGC -A AGATCGGAAGAGC –minimum-length 30 -q 20,20'. Gene-level abundance estimates were generated from the trimmed reads using RSEM v1.2.31 (*Li and Dewey, 2011*) running STAR v2.5.1b

(*Dobin et al., 2013*) with default settings, aligned against the *Mus musculus* Ensembl release 89 transcriptome (mm10). All further analysis was conducted using the DESeq2 (*Love et al., 2014*) package from Bioconductor v3.5 run in R v3.4.0. The expected counts were imported and rounded to integers to generate a counts matrix. Differential expression between phenotype groups was assessed using the DESeq function with default settings. In the case of comparisons of different MSP1$_{21}$-specific B cell subsets obtained from the same experimental mouse, an additional mouse factor was added to the design formula to accommodate the paired nature of the data. Significance was thresholded using an FDR $\leq$ 0.01. PCA analysis was conducted using DESeq's plot PCA function with the regularized log (rlog) transformed count data. Heat maps were generated using the regularized log (rlog) transformed count data, scaled per gene using a z-score. Mouse homologues to genes previously associated with human AMB were selected (*Supplementary file 1*), and those showing significant differential expression on MSP1$_{21}$-specific AMB were used to produce separate heat maps split by functional annotation (*Figure 3*). The GSEA pre-ranked function from the Broad's Gene Set Enrichment Analysis (GSEA) (*Subramanian et al., 2005*) suite was used to assess significant enrichment of MSigDB's C2 Reactome gene sets associated with differential expression between cell types. The function was run using a list of genes ranked for differential expression using DESeq2's Wald test statistic with default settings except for: collapse dataset to gene = false enrichment statistic = classic

## Statistical analysis

Statistical analysis was performed using Mann Whitney U test, Kruskal-Wallis test followed by Dunn's multiple comparisons test, or Two-Way ANOVA followed by Dunnett's multiple comparisons test on Prism software version 6 (GraphPad). $p < 0.05$ was accepted as a statistically significant difference.

## Acknowledgements

We are grateful to Graham Preece, Philip Hobson and the Flow Cytometry Facility, Jackie Holland and the Biological Research Facility, Richard Mitter and the Bioinformatics Facility, and the Advanced Sequencing Facility at the Francis Crick Institute and the former MRC National Institute for Medical Research, London, UK. We thank Robert Brink, Garvan Institute of Medical Research, New South Wales, Australia, for donating the the C57Bl/6 IgH HEL variable region knock-in construct; Marion Pepper, University of Washington, for her support with the preparation of the MSP1$_{21}$ tetramers; and Dinis Calado and Matthew Lewis, Francis Crick Institute, for their critical reading of the manuscript.

This work was supported by the Francis Crick Institute which receives its core funding from the UK Medical Research Council (FC001101), Cancer Research UK (FC001101) and the Wellcome Trust (FC001101); by the Wellcome Trust (grant reference WT101777MA). Irene Tumwine is the recipient of a Francis Crick PhD studentship.

## Additional information

### Funding

| Funder | Grant reference number | Author |
|---|---|---|
| National Institutes of Health | AI110553 | Randall S Davis |
| Medical Research Council | FC001101 and FC001194 | Victor LJ Tybulewicz |
| Cancer Research UK | FC001101 and FC001194 | Victor LJ Tybulewicz |
| Wellcome Trust | FC001101 and FC001194 | Victor LJ Tybulewicz |
| Medical Research Council | FC001101 | Jean Langhorne |
| Wellcome Trust | FC001101 | Jean Langhorne |
| Cancer Research UK | FC001101 | Jean Langhorne |

The funders had no role in study design, data collection and interpretation, or the decision to submit the work for publication.

## Author contributions
Damián Pérez-Mazliah, Conceptualization, Data curation, Formal analysis, Validation, Investigation, Visualization, Methodology, Writing—original draft; Peter J Gardner, Conceptualization, Formal analysis, Validation, Investigation, Methodology, Writing—review and editing; Edina Schweighoffer, Conceptualization, Formal analysis, Supervision, Investigation, Methodology, Writing—review and editing; Sarah McLaughlin, Caroline Hosking, Irene Tumwine, Investigation, Writing—review and editing; Randall S Davis, Resources, Writing—review and editing; Alexandre J Potocnik, Conceptualization, Supervision, Methodology, Writing—review and editing; Victor LJ Tybulewicz, Conceptualization, Supervision, Validation, Methodology, Funding acquisition, Writing—review and editing; Jean Langhorne, Conceptualization, Supervision, Funding acquisition, Validation, Methodology, Project administration, Writing—review and editing

## Author ORCIDs
Damián Pérez-Mazliah (ID) http://orcid.org/0000-0002-2156-2585
Peter J Gardner (ID) http://orcid.org/0000-0002-4940-2639
Randall S Davis (ID) http://orcid.org/0000-0003-1906-8219
Alexandre J Potocnik (ID) http://orcid.org/0000-0002-4730-8593
Victor LJ Tybulewicz (ID) https://orcid.org/0000-0003-2439-0798
Jean Langhorne (ID) https://orcid.org/0000-0002-2257-9733

## Ethics
Animal experimentation: The study was carried out in accordance with the UK Animals (Scientific Procedures) Act 1986, Home Office license 80/2538 and 70/8326. Was approved by the MRC National Institute for Medical Research Ethical Committee, The Francis Crick Institute Ethical Committee, and further approved by the Home Office of the UK upon granting of the HO license.

## Decision letter and Author response
Decision letter https://doi.org/10.7554/eLife.39800.022
Author response https://doi.org/10.7554/eLife.39800.023

# Additional files
## Supplementary files
• Supplementary file 1. Mouse homologues to human genes previously described to be either up (↑) or down (↓) regulated in human atypical memory B cells (AMB).
DOI: https://doi.org/10.7554/eLife.39800.016

• Supplementary file 2. List of top 50 Reactome gene sets yielding the highest normalized enrichment score (NES) by GSEA.
DOI: https://doi.org/10.7554/eLife.39800.017

• Transparent reporting form
DOI: https://doi.org/10.7554/eLife.39800.018

## Data availability
RNAseq transcriptome data have been deposited in GEO under accession code GSE115155.

The following dataset was generated:

| Author(s) | Year | Dataset title | Dataset URL | Database and Identifier |
|---|---|---|---|---|
| Pérez-Mazliah D, Gardner PJ, Schweighoffer E, McLaughlin S, Hosking C, Tumwine I, Davis RS, Potocnik A, Tybu- | 2018 | Plasmodium-specific atypical memory B cells are short-lived activated B cells | https://www.ncbi.nlm.nih.gov/geo/query/acc.cgi?acc=GSE115155 | NCBI Gene Expression Omnibus, GSE115155 |

lewicz V, Langhorne
J

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
