## [Decision Letter]

Thank you for submitting your article "*Plasmodium*-specific atypical memory B cells are short-lived activated B cells" for consideration by *eLife*. Your article has been reviewed by Michel Nussenzweig as the Senior Editor, a Reviewing Editor, and three reviewers. The reviewers have opted to remain anonymous.

The reviewers have discussed the reviews with one another and the Reviewing Editor has drafted this decision to help you prepare a revised submission.

Essential revisions:

All three reviewers agreed that the study is of sufficient general interest but that the manuscript requires significant re-organization and rewriting to clarify the various points that they have collectively raised. For this, I am copying the reviewers’ comments below. While this will not require new experimentation, it will be important that you revise the manuscript to incorporate and address all reviewers concerns.

Reviewer #2:

Atypical memory B cells (AMB) are a feature of *Plasmodium* (malaria) infections and B cells with a similar phenotype (CD21^lo^, FCRL^hi^ etc.) have been identified in many other human infections and immuno-deficiencies as well as in aging mice. There is still much conjecture around the function and of AMBs and what triggers their development, partly because they are frequently described in the context of active human infections. Studies in which the genesis and fate of AMBs can be studied in a more controlled manner are likely to provide greater insight into the nature of these cells.

In this manuscript, the authors have developed a novel BCR/Ig knock-in mouse in which B cells express a heavy chain recognising the *P. chabaudi* Merozoite Surface Protein 1 (MSP1) antigen allowing direct analysis of the response of anti-*Plasmodium* B cells. Importantly, responses were triggered in chimeric mice carrying low frequencies (~1%) of anti-MSP1 B cells and using the natural route of *P. chabaudi* infection i.e. via mosquito bite. Thus, the authors have utilised a carefully designed, relevant and informative model for this study.

The authors demonstrate that a population of B cells with an AMB-phenotype is generated but that this population does not persist long-term. It would be important to confirm that these are not GC B cells (see point 3, below). It is also apparent that an FCRL5^hi^ population of conventional memory B cells does persist, providing new information about AMBs in the context of *Plasmodium* infection. Whilst this main result is an important one, there are a number of issues with the study that do need to be addressed.

1) MSP1_21_ needs to be defined (the 21kDa C-terminal fragment of MSP1).

2) For clarity it should be pointed out that a recombinant light chain is not required and that most endogenous light chains will pair with the NIMP23 heavy chain to generate a BCR with detectable binding to MSP1. Can any assessment be made of the affinity cut-off for detection with the MSP1 fluorescent probes?

3) Please explain briefly in the Results section the nature of the MSP1 fluorescent probes.

4) It is unusual that CD2 has been used to delineate early B-lineage subsets in Figure 1A. Please provide a reference verifying this marker can be used for this purpose.

5) The decision to separate the anti-MSP1 B cells present on day 35 of the response into 4 populations based solely on CD11b and CD11c staining is confusing. Although around two-thirds of the cells are GC B cells at this time point (Figure 1—figure supplement 2G), it appears that no attempt has been made to exclude GC B cells from the analyses in Figure 2 or the sorting/mRNAseq analysis shown in Figure 3. Why was this not done? Although the AMBs (CD11b^+^, CD11c^+^) are presumably not GC B cells (this needs to be clarified), the "non-AMB" CD11b^-^, CD11c^-^ population seems likely to contain many GC B cells as well as potentially other memory B cells. It is unclear how the mRNAseq data are to be meaningfully interpreted in this case. The fact that GC B cells are included in these analyses needs to be clearly indicated and the impact of this on the interpretation of the data spelled out. In particular, it is difficult to see how any meaningful comparison can be made between the gene expression profiles of AMBs and "non-AMBs".

6) Please explain more clearly the justification of the genes chosen to display in Figure 3. In the Materials and methods section it mentioned that "top 50 most up and down regulated gene heat maps" were produced and then that "Significant selected marker genes were used to produce separate heat maps split by functional annotation." Are the "Top 50" gene sets shown anywhere? And are the "heat maps split by functional annotation" shown in Figure 3 a subset of those in the "Top 50" gene sets? If the answer to both these questions is no, then the reference to these "Top 50" sets should be removed.

8) In addition, an indication of what the units on the heat map scale are is required. Is this a z-score or log2-fold change or something else?

7) The data in Figure 4 showing a transient population of CD11b^+^ CD11c^+^ B cells appearing 1 day after challenge with MSP1 protein is of dubious relevance to the generation/survival of AMBs. There is no way in which these cells could be described as any type of memory B cell. These data are misleading and should be removed.

8) In the legend for Figure 5I, the first "MSP1_21_-specific B cells" should be removed.

9) The lengthy discussion about FCRL5 being used to identify the MSP1_21_-specific B_mem_ cells was somewhat distracting/confusing in relation to that study's focus on AMBs. It subsequently became clear that it was the marker used to sort the B_mem_. As it is not a well described marker for memory B cells it did need to be justified but an addition to line 325 at the end of FCRL5 Results section, "… FCLR5 identifies *P. chabaudi*-specific B_mem_" indicating that it was subsequently used as a marker to sort on the B_mem_ cells, would make the reason for the focus on FCRL5 more clear.

10) In subsection “*Plasmodium*-specific B_mem_ express high levels of FCRL5” and subsection “Mice” the authors refer to FCRL5 (and CD80) as markers of "antigen-experienced B cells." Since GC B cells and plasma cells are also antigen experienced B cells this statement needs to be made less general so as it applies to just AMBs and B_mem_ cells.

*Reviewer #3:*

The authors generate ki mice that express the heavy chain gene of a previously reported hybridma antibody with reactivity to the *Plasmodium chabaudi* blood stage surface protein, MSP1. Surprisingly, 60% of all B cells in this model bind MSP1 with only the heavy chain ki allele. B cell numbers appear normal, but the high frequency of MSP1-reactive B cells suggests that the heavy chain alone mediates antigen binding. A much less likely explanation may be that the repertoire is biased towards usage of a restricted light chain set and therefore might be highly clonal.

Unfortunately, I cannot find information on the gene and SHM load, but it is likely mutated. It is also unclear how strongly the naïve cells react to MSP1. A better description and discussion of the ki model is necessary to allow the reader to better judge the findings related to the development of AMBs and to support the physiological relevance of the model and findings. One way of demonstrating this may be to assess the response of the non-ki cells in the transfer model.

The authors state that they developed the model to study the formation of AMBs in response to mosquito-mediated *P. chabaudi* infection. AMBs have been described in humans. There is a debate as to whether AMBs represent exhausted/non-functional cells or whether they actively contribute to the immune response. In malaria, the latter is supported by the fact that they show signs of proliferation, CSR, and antibody secretion, but definite proof is difficult to obtain in humans.

Using a transfer model of the MSP1-specific B cells to obtain more physiological numbers of antigen-specific naïve B cells, the authors demonstrate that a MSP1-specific B cell population resembling human AMBs transiently develops in their model and also after MSP1 immunization. The evidence is based on thorough phenotypic and transcriptome analyses, which reveal a plasmablast-like signature suggesting that the cells are activated, short-lived B cells that actively participate in the immune response. Classical memory B cells are also generated in the model and persist after resolution of the infection. These cells express high levels of FCRL5, which established FCRL5 as a marker for memory B cells in this model.

Technically, this is a very nice study, which demonstrates that AMBc develop also in a mouse model of malaria after mosquito transmission. The findings provide additional support for the notion that these cells actively participate in the response, but unfortunately there is no direct evidence. Do these cells secrete antibodies against the parasite? What about the phenotype of the endogenous B cells in the model? Is the development of AMBs linked to the chronicity of the response and blood stage parasitemia independent of the antigen or ki gene and presumably high affinity of the antibodies? What is the cellular origin of the cells? Do they develop in GCs?

It is not relevant to answer all questions in the manuscript, but some clear evidence for the function/role of AMBs should be provided to advance over previous descriptive studies in humans. The model should allow the authors to address these important questions.

Reviewer #4:

In this manuscript the authors present a rigorous analysis of the impact of *P. chabaudi* infection in mice on antigen-specific (MSP1_21_-specific) B cells from IgH knock-in mice. The authors interpret their data in the context of the documented expansion in humans of what have been termed atypical memory B cells abbreviated here as AMBs. Indeed, defining AMBs in mice as CD11b^+^ CD11c^+^ FcRL5^+^ CD21LO the authors observe increases in the numbers of MSP1_21_-specific B cells with these markers following infection and claim that the infection generated antigen-specific human-like AMB. The problem is that in mice (but not in humans) FcRL5 expression is a discrete/unique marker of marginal zone B cells and B-1a and B-1b cells (often referred to as innate-like B cells) (Davis, (2015); Won et al., (2006)). Thus, by Occam's razor it's fair to assume that authors are describing the impact of infection on marginal zone and B1 B cells. The authors only comment on the expression of FcRL5 in marginal B cells obliquely in the Discussion section.

I believe it is fair to say that the relationship, if any, between B cells in mice that express T-bet (termed age-related B cells or ABCs) and human atypical B cells (also referred to as tissue-like memory B cells) is not established (Portugal et al., 2017), nor are the identities of unusual B cell populations between all human chronic infections and in autoimmunity. Thus, it would seem to be premature to relate the findings here to an aggregate B cell phenotype in humans particularly when the authors have chosen a discrete/unique marker of mouse innate-like B cells to define AMB. I believe the data here merit publication but as a study of marginal zone and B1 B cells not as a model of human atypical memory B cells.

---

## [Author Response]

Reviewer #2:[…] 1) MSP1_21_ needs to be defined (the 21kDa C-terminal fragment of MSP1).

We have corrected this, please see the Introduction.

2) For clarity it should be pointed out that a recombinant light chain is not required and that most endogenous light chains will pair with the NIMP23 heavy chain to generate a BCR with detectable binding to MSP1. Can any assessment be made of the affinity cut-off for detection with the MSP1 fluorescent probes?

This statement has been added to the text, subsection “Generation of an immunoglobulin heavy chain knock-in transgenic mouse model to study *Plasmodium*-specific B cell responses.

3) Please explain briefly in the Results section the nature of the MSP1 fluorescent probes.

A brief explanation of the probe has now been added to the Results section.

4) It is unusual that CD2 has been used to delineate early B-lineage subsets in Figure 1A. Please provide a reference verifying this marker can be used for this purpose.

The following references have been added to the Results section and to the reference list:

Sen, Rosenberg and Burakoff 1990.

Young et al., 1994

5) The decision to separate the anti-MSP1 B cells present on day 35 of the response into 4 populations based solely on CD11b and CD11c staining is confusing. Although around two-thirds of the cells are GC B cells at this time point (Figure 1—figure supplement 2G), it appears that no attempt has been made to exclude GC B cells from the analyses in Figure 2 or the sorting/mRNAseq analysis shown in Figure 3. Why was this not done? Although the AMBs (CD11b^+^, CD11c^+^) are presumably not GC B cells (this needs to be clarified), the "non-AMB" CD11b^-^, CD11c^-^ population seems likely to contain many GC B cells as well as potentially other memory B cells. It is unclear how the mRNAseq data are to be meaningfully interpreted in this case. The fact that GC B cells are included in these analyses needs to be clearly indicated and the impact of this on the interpretation of the data spelled out. In particular, it is difficult to see how any meaningful comparison can be made between the gene expression profiles of AMBs and "non-AMBs".

We agree with the reviewer that the non-AMB CD11b^-^CD11c^-^ at day 35 π represents a complex population very likely including different B-cell subsets. However, our interest rather lies on the AMB CD11b^+^CD11c^+^. We have shown by the PCA analysis (Figure 7), this is clearly a distinctive subset of B cells, strikingly different from all the other subsets analysed in this study, including non-AMB CD11b^-^CD11c^-^ as well as conventional B_mem_ and naïve B cells. Figure 3K shows that MSP1-specific AMB CD11b^+^CD11c^+^ have plasmablasts/plasma cells characteristics, rather than GC. Moreover, as shown in Figure 5G-I, while the majority of MSP1-specific AMB CD11b^+^CD11c^+^ express CD80, and, in lower proportion, CD273, MSP1-specific CD38^lo^GL7^hi^ GC B cells do not express any of these two memory markers.

In order to clarify this point better, we have now added an additional Figure 8 to the manuscript, to provide further details of the phenotypical characteristics of MSP1-specific AMB CD11b^+^CD11c^+^. Figure 8C shows that GC B cells (CD38^lo^GL-7^hi^) are contained within the non-AMB CD11b^-^CD11c^-^, while the MSP1-specific AMB CD11b^+^CD11c^+^ show no GC surface markers, clearly showing that these AMB are not GC B cells. Moreover, the MSP1-specific GC B cell pool shows low expression of CD11b and CD11c, clearly setting them apart from the MSP1-specific AMB CD11b^+^CD11c^+^ subset (Figure 8E). This is summarised in subsection “*Plasmodium*-specific AMB are a distinct short-lived activated B cell subset” and discussed in the Discussion section.

Therefore, we believe this is sufficient evidence to demonstrate that MSP1-specific AMB CD11b^+^CD11c^+^ represent a B-cell subset distinct from GC.

Nonetheless, we cannot rule out that the MSP1-specific AMB CD11b^+^CD11c^+^ might arise from the GC, or even represent a late GC stage/early GC emigrant. This will be the focus of future more extensive studies.

6) Please explain more clearly the justification of the genes chosen to display in Figure 3. In the Materials and methods section it mentioned that " top 50 most up and down regulated gene heat maps" were produced and then that " Significant selected marker genes were used to produce separate heat maps split by functional annotation." Are the "Top 50" gene sets shown anywhere? And are the " heat maps split by functional annotation" shown in Figure 3 a subset of those in the "Top 50" gene sets? If the answer to both these questions is no, then the reference to these "Top 50" sets should be removed.

The "Top 50" genes was erroneously carried over from older versions of the manuscript. This has now been corrected and removed from the revised manuscript.

The genes in Figure 3 were chosen based on previous human AMB reports (Supplementary file 1). These genes were previously shown to be either up or downregulated on human AMB. Those showing significant differential expression on MSP1_21_-specific AMB were used to produce separate heatmaps split by functional annotation. This gene selection to produce Figure 3 was further clarified in the Results section, and the Materials and methods section.

8) In addition, an indication of what the units on the heat map scale are is required. Is this a z-score or log2-fold change or something else?

As stated in Materials and methods section “Heatmaps were generated using the regularized log (rlog) transformed count data, scaled per gene using a z-score”.

The units given in the heatmaps are z-scores of gene abundance, which have the effect of zero-centering the relative change across samples and aids visualisation.

7) The data in Figure 4 showing a transient population of CD11b^+^ CD11c^+^ B cells appearing 1 day after challenge with MSP1 protein is of dubious relevance to the generation/survival of AMBs. There is no way in which these cells could be described as any type of memory B cell. These data are misleading and should be removed.

Data in Figure 4 shows that MSP1-specific CD11b^+^CD11c^+^ AMB cells are induced in response to immunizations. As stated by the reviewer, these cells are short-lived with no functional characteristics of memory B cells. Our argument is that mouse (and quite possibly human) atypical memory B cells are not really “memory” cells, but rather short-lived activated B cells. Therefore, data in Figure 4 provides further support to our argument.

Also, this result demonstrates that the presence of the pathogen is not required to activate these cells, as a *P. chabaudi*-derived antigen is sufficient to generate them in the context of a specific immunization strategy. Thus, it appears that MSP1-specific CD11b^+^CD11c^+^ AMB cells are not a result of aberrant B-cell activation driven by the pathogen, but rather a normal component of the B-cell response in the context of specific signals.

We have now changed the manuscript to try to emphasize and clarify these points, subsection “Generation of *Plasmodium*-specific AMB in response to immunization”.

8) In the legend for Figure 5I, the first "MSP1_21_-specific B cells" should be removed.

This has been corrected.

9) The lengthy discussion about FCRL5 being used to identify the MSP1_21_-specific B_mem_ cells was somewhat distracting/confusing in relation to that study's focus on AMBs. It subsequently became clear that it was the marker used to sort the B_mem_. As it is not a well described marker for memory B cells it did need to be justified but an addition to line 325 at the end of FCRL5 Results section, "… FCLR5 identifies P. chabaudi-specific B_mem_" indicating that it was subsequently used as a marker to sort on the B_mem_ cells, would make the reason for the focus on FCRL5 more clear.

The discussion related to FCRL5 as a memory B cell marker has been reduced.

10) In subsection “Plasmodium-specific B_mem_ express high levels of FCRL5” and subsection “Mice” the authors refer to FCRL5 (and CD80) as markers of "antigen-experienced B cells." Since GC B cells and plasma cells are also antigen experienced B cells this statement needs to be made less general so as it applies to just AMBs and B_mem_ cells.

This phrase has now been removed.

Reviewer #3:The authors generate ki mice that express the heavy chain gene of a previously reported hybridma antibody with reactivity to the Plasmodium chabaudi blood stage surface protein, MSP1. Surprisingly, 60% of all B cells in this model bind MSP1 with only the heavy chain ki allele. B cell numbers appear normal, but the high frequency of MSP1-reactive B cells suggests that the heavy chain alone mediates antigen binding. A much less likely explanation may be that the repertoire is biased towards usage of a restricted light chain set and therefore might be highly clonal.Unfortunately, I cannot find information on the gene and SHM load, but it is likely mutated.

The degree of SHM in the transgenic heavy chain, which is demonstrated in Figure 1—figure supplement 1, is approximately 88 to 90% of germline. This level of SHM suggests that this particular HC derives from a B cell that underwent affinity maturation by passaging through the GC.

It is also unclear how strongly the naïve cells react to MSP1. A better description and discussion of the ki model is necessary to allow the reader to better judge the findings related to the development of AMBs and to support the physiological relevance of the model and findings. One way of demonstrating this may be to assess the response of the non-ki cells in the transfer model.

As shown in Figure 1E, the frequency of non-Tg B cells that react with the MSP1 probe is at least 100-fold lower than the frequency of Tg B cells that react to the probe. Thus, we interpret the signal from the non-Tg B cells as background. Importantly, this does not change after adoptive transference (Figure 1—figure supplement 1D) and infection-driven activation (Author response image 1).

[…] The findings provide additional support for the notion that these cells actively participate in the response, but unfortunately there is no direct evidence. Do these cells secrete antibodies against the parasite? What about the phenotype of the endogenous B cells in the model? Is the development of AMBs linked to the chronicity of the response and blood stage parasitemia independent of the antigen or ki gene and presumably high affinity of the antibodies? What is the cellular origin of the cells? Do they develop in GCs?It is not relevant to answer all questions in the manuscript, but some clear evidence for the function/role of AMBs should be provided to advance over previous descriptive studies in humans. The model should allow the authors to address these important questions.

We thank the reviewer for the questions raised from the study. We agree that all of these important questions should be answered. This indeed is the purpose of developing the model. However, our manuscript already contains a vast amount of data, and highlights important points about the characteristics of these cells in this model. The questions posed would constitute a large number of very long-term experiments to answer these points satisfactorily. They will be the subject of a further manuscript. However, we have raised them as important follow-up experiments in the Discussion section.

Reviewer #4:[…] The problem is that in mice (but not in humans) FcRL5 expression is a discrete/unique marker of marginal zone B cells and B-1a and B-1b cells (often referred to as innate-like B cells) (Davis, (2015); Won et al., (2006)). Thus, by Occam's razor it's fair to assume that authors are describing the impact of infection on marginal zone and B1 B cells. The authors only comment on the expression of FcRL5 in marginal B cells obliquely in the discussion.

Please, see answer to question 2 below.

I believe it is fair to say that the relationship, if any, between B cells in mice that express T-bet (termed age-related B cells or ABCs) and human atypical B cells (also referred to as tissue-like memory B cells) is not established (Portugal et al., 2017), nor are the identities of unusual B cell populations between all human chronic infections and in autoimmunity. Thus, it would seem to be premature to relate the findings here to an aggregate B cell phenotype in humans particularly when the authors have chosen a discrete/unique marker of mouse innate-like B cells to define AMB. I believe the data here merit publication but as a study of marginal zone and B1 B cells not as a model of human atypical memory B cells.

The initial work characterizing the receptor distribution of FCRL5 (Won et al.,. 2006), was performed on tissues from mice at homeostasis. Up to this point, we are not aware of any published studies performed on immune challenged mice for FCRL5 protein expression by memory B cells. In unpublished data, generated in an independent laboratory from the senior author, we have identified small numbers of class-switched (CS) cells in the spleens of unchallenged mice that lack typical MZ markers. In these separate studies, FCRL5 surface expression overlaps with the expression of memory markers identified by Shlomchik as detailed in the current manuscript. Furthermore, in other preliminary studies, FCRL5 expression also appears to be expressed by class-switched B cells in mice following NP-CGG challenge. Thus, rather than a molecule restricted to innate-like B cells, FCRL5 may serve as a marker/regulator of B lineage cells situated at a peak stage of maturity that are poised for antigen reactivity and terminal differentiation (i.e. Ab secretion).

Moreover, in order to clarify this point further, we have now added to the manuscript an additional Figure 8 to provide further details of the phenotypical characteristics of MSP1-specific AMB CD11b^+^CD11c^+^. As discussed elsewhere (Baumgarth,2010, Garraud et al., 2012, Zouali and Richard 2011, Pillai et al., 2005), B1 and MZ B cells are CD1d^mid/hi^, CD9^+^, IgM^hi^ and CD23^—^. B1 are also CD43^+^, B220^lo^, and may (B1a) or may not (B1b) express CD5. MZ are further characterised by CD22^hi^, CD21/CR2^hi^, and the expression of the lysophospholipid sphingosine-1 phosphate receptor S1P_1_, and the lineage master regulator *Notch2*. Indeed, MSP1_21_-specific CD11b^+^CD11c^+^ AMB showed high expression of some markers associated with B1 and/or MZ, including CD5, CD9 and CD43. However, these markers have also been shown to be highly expressed by activated B2 B cells and/or plasma cells (Baumgarth,2010). Studying all these canonical markers, we found no other similarities between B1/MZ and MSP1_21_-specific CD11b^+^CD11c^+^ AMB to suggest a relationship between these subsets (Figure 8A). This was further corroborated by flow cytometry analysis (Figure 8B). Moreover, unlike MZ and B1 B cells, MSP1_21_-specific AMB were class-switched and expressed very high levels of IgG (Figure 7D). Nonetheless, B1 B cells have been shown to class-switch and contribute to serum IgG1, IgG2a and IgA to influenza (Baumgarth, 2005), and IgG-producing B1a B cells have been shown to accumulate in the spleen of a mouse model of systemic lupus erythematosus (Enghard, 2010). Therefore, we cannot absolutely rule out the possibility that AMB might represent a particular B1 B cell subset that expands in the spleen and blood in response to *Plasmodium* infection. However, it is important to highlight that in these previous studies (Baumgarth, 2005; Enghard, 2010) the B1a compartment producing IgG was no more than 20% of the total B1a compartment (the other 80% producing IgM). As shown in Figure 8B and Figure 7, the MSP1-specific AMB identified here are IgM^lo^ and express very high levels of IgG. This is unlike any other B1 B cells described so far, and rather suggest a GC origin.

We hope we have clarified this in the text, subsection “*Plasmodium*-specific AMB are a distinct short-lived activated B cell subset” and the Discussion section.

We used publicly available RNA expression data from an extensive series of experiments defining AMB in humans (summarized in Supplementary file 1) to determine whether these mouse AMB cells have gene expression profiles similar to the human AMB cells. It is true that our FACS analyses on MSP1-specific AMB included a limited number of surface markers based on previous data generated in humans (i.e. CD11b, CD11c, CD80, IgD, IgM, FCRL-5 and CD21). Nonetheless, we performed extensive transcriptome analysis on sorted MSP1-specific AMB. Our conclusion that the cells we identify here resemble human AMB is based on an extensive and deep comparison of the transcriptome of the MSP1-specific AMB identified here with human AMB identified in different infections, including *Plasmodium* and HIV (Please, refer to Supplementary file 1 and Figure 3). Therefore, we do not agree with the reviewer that we arrive at this conclusion only based on the expression of a few surface marker, and believe our analysis is much more robust and that the similarities between MSP1-specific AMB and human AMB are indeed striking.